

# Examining the role of varying surface pressure in the climate of early Earth

Junyan Xiong[1] and Jun Yang[1]

[1]Department of Atmospheric and Oceanic Sciences, School of Physics, Peking University, Beijing 100871, China.

**Correspondence:** Jun Yang (junyang@pku.edu.cn)

**Abstract.**

During the Archean Eon in 2.7 billion years ago, solar luminosity was about 75% of the present-day level, but the surface temperature was suggested to similar to or even higher than modern. What mechanisms act to maintain the temperate climate of early Earth is not clearly known yet. Recent studies suggested that surface air pressure was different from the present level. How does varying surface air pressure influence the climate? Using an atmospheric general circulation model coupled to a slab ocean with specified oceanic heat transport, we show that decreasing (increasing) surface pressure acts to cool (warm) the surface mainly because the greenhouse effect of pressure broadening becomes weaker (stronger). The effect of halfing or doubling the surface pressure on the global-mean surface temperature is about 10 K or even larger when ice albedo feedback or water vapor feedback is strong. If the surface pressure was 0.5 bar, a combination of a $CO_2$ partial pressure of about 0.04 bar and an oceanic heat transport of twice the present-day level or a combination of a $CO_2$ partial pressure of about 0.10 bar and an oceanic heat transport of half the present-day level is required to maintain a climate similar to modern, under a given $CH_4$ partial pressure of 1 mbar. Future work with fully coupled atmosphere-ocean models is required to explore the strength of oceanic heat transport and with cloud resolving models to examine the strength of cloud radiative effect under different surface air pressures.

## 1 Introduction

During the Archean Era, the solar luminosity was about 75% of the present day (Gough, 1981; Blake et al., 2010). If other climate parameters were the same as their present values, Earth would have been much colder and entirely glaciated during its early period (Sagan and Mullen, 1972). However, geochemical proxies suggest that the tropical ocean of early Earth in 2.9 billion years ago (Ga) might be ice-free and the climate was similar to or even warmer than today (Hren et al., 2009). The contrast between paleoclimate proxies and climate theory is known as the 'faint young Sun paradox'. To hold the hospitable Archean climate, a stronger greenhouse effect was required (Owen et al., 1979; Walker et al., 1981; Pavlov et al., 2000).

Archean temperature can be estimated from isotopic determination of marine sediments. Oxygen isotopic composition between seawater and sediments was used as an indication of temperature, and a low-$\delta^{18}$O sediment infer a high ocean temperature (Kasting and Howard, 2006). The seawater temperature was inferred to reach 330–360 K from oxygen isotope data (Knauth and Lowe, 2003). Observed silicon isotope also implies a high seawater temperature of about 340 K in 3.5 Ga (Robert



and Chaussidon, 2006). However, the observed low-$\delta^{18}$O sediment might be caused by local geothermal heat flows, therefore the temperatures inferred are not the representation of global mean (Shields and Kasting, 2007; van den Boorn et al., 2007; Feulner, 2012). Other studies suggested a moderate temperature, no higher than 313 K during the mid-Archean; a temperate Archean climate is more acceptable (Hren et al., 2009; Blake et al., 2010).

It is generally validated that ancient $CO_2$ concentration was much higher than the present level. On geological timescale, $CO_2$ is released to the atmosphere through volcanoes and metamorphism and moved from the atmosphere by weathering of silicate minerals (Walker et al., 1981). A lower surface temperature leads to weaker weathering of silicate minerals, therefore the residence time of $CO_2$ in the atmosphere would be longer. A longer residence time is conducive to accumulation of $CO_2$ in the air. As a result, a lower surface temperature tends to keep a higher $CO_2$ partial pressure, which warms the planet, and

vice versa, forming a negative feedback for the long-time evolution of Earth's climate. Analyses of ferrous carbonate minerals suggested that the $CO_2$ partial pressure in 3 Ga was about 0.0025 to 0.04 bar, about 8-145 times of the pre-industrial level (Rye et al., 1995; Hessler et al., 2004). Driese et al. (2011) suggested that the $CO_2$ partial pressure was 10-50 times of pre-industrial level in 2.69 Ga based on the weathering of the Saganaga Tonalite.

  Methane likely played an important role in the Archean climate (Haqq-Misra et al., 2008). Methanogenesis is an anaero-

bic process used by methanogens to produce energy. This process releases methane into the surrounding environment from $H_3C\text{-}COOH$. Meanwhile, the major removal mechanism of methane is oxidation by hydroxyl radical. Because the Archean atmosphere is anoxic, the removal process of methane from the atmosphere would be much slower than present. Therefore, the residence time of methane would be longer, leading to a methane-abundant atmosphere. However, if methane concentration was so high that an organic haze can form when the ratio of $CH_4$ to $CO_2$ is larger than 0.2 (Trainer et al., 2006), there would

be a cooling effect because of stronger scattering and less shortwave flux reaching the surface. As Earth's surface methanogenesis is positively correlated with surface temperature, there is an upper limit on $CH_4$ concentration (Pavlov et al., 2001). The methane flux could hold a methane mixing ratio of about $10^{-3}$ (Pavlov et al., 2000).

  $N_2$ and $O_2$ are the key bulk atmospheric constituents of the present air. During the early Earth period, $O_2$ partial pressure was lower than 1% of the pre-industrial level before the first Great Oxidation Event (GOE) in about 2.45 Ga (Bekker et al.,

2004; Canfield, 2005). Recent studies shown that $N_2$ pressure was quite different from present, but whether it is lower or higher than present is not constrained yet. One method is measuring the ratio of $N_2$ to Ar, with assuming atmospheric $^{36}$Ar concentration to be constant since 3.5 Ga. Using this method, the study of Marty et al. (2013) and Avice et al. (2018) suggested that the upper limit of Archean $N_2$ partial pressure is likely lower than 1.1 bar, maybe as low as 0.5 bar in 3.0 to 3.5 Ga. Fossil raindrop impressions were also used to estimate the surface air density. When assuming the maximum raindrop diameters

were essentially identical to today's, an empirical relation between air density and maximal terminal velocity, $V_{term} \propto \rho_{air}^{-1/2}$, was used to estimate the surface air density (Som et al., 2012). The surface air pressure was constrained to be lower than present. However, this method has large uncertainties mainly because raindrop imprint size distribution depends more strongly on rainfall rate instead of surface air density (Kavanagh and Goldblatt, 2015). The upper limit on Archean surface air pressure could be as high as about 9 bar based on the maximum raindrop size (Kavanagh and Goldblatt, 2015) or 3 bar based on the

bulk silicate Earth nitrogen inventory (Goldblatt et al., 2009; Mallik et al., 2018; Johnson and Goldblatt, 2015). Differences in





the gas bubble size between the top of basaltic lava flows and the bottom of the flows suggested that the air pressure in 2.7 Ga might be as low as $0.23 \pm 0.23$ bar (Som et al., 2016).

Background atmospheric pressure doesn't cause direct greenhouse effect, nevertheless, it can influence the climate via various ways. (1) Air pressure influences shortwave radiation via Rayleigh scattering. Dense atmosphere reflects more shortwave radiation and increases planetary albedo, thus less shortwave flux reaches the surface (Hartmann, 2016). (2) Air pressure influences the thermal absorption of greenhouse gases. Pressure broadening widens the spectral lines of greenhouse gases such as $H_2O$, $CO_2$, and $CH_4$, leading to a positive radiative effect (Goldblatt et al., 2009; Wolf, 2014). Meanwhile, collision-induced continuum absorption, such as the collision pairs of $N_2$-$N_2$, $N_2$-$CH_4$ and $N_2$-$H_2$, warms the surface if the atmospheric pressure is high (Wordsworth and Pierrehumbert, 2013; Pierrehumbert, 2010). (3) Different background atmospheric pressure leads to different molecular weights and heat capacities. Using a three-dimensional (3-D) idealized GCM, Kaspi and Showman (2015), Chemke et al. (2016), and Chemke and Kaspi (2017) found that an increase in the atmospheric heat capacity with increasing air mass decreases the net radiative cooling in the lower layers of the atmosphere, which trends to warm the surface; moreover, vertical heat advection by eddies decreases with increasing air pressure, which further warms the surface; both these two effects are more effective in middle and high latitudes. Therefore, the meridional temperature gradient decreases with increasing air pressure, and the tropical trade winds and extratropical eddies and jets become weaker in strength and smaller in length scales. The reduced horizontal surface temperature gradient with air pressure was also confirmed in the simulations with other global models (Komacek and Abbot, 2019) and even on tidally locked exoplanets (such as Yang et al. (2019)). (4) Background air pressure also influences the thermal stratification of the atmosphere through its effect on the moist adiabatic lapse rate, which is given by $g\frac{R_{sd}T^2+L_v rT}{c_{pd}R_{sd}T^2+L_v^2 r\epsilon}$, where $g$ is gravitational acceleration, $L_v$ is the heat of water vapor condensation, $R_{sd}$ is the specific gas constant of dry air, $r$ is the ratio of the mass of water vapor to the mass of dry air, and $\epsilon$ is the ratio of the specific gas constant for dry air to the specific gas constant for water vapor (Stone and Carlson, 1979; Charnay et al., 2013). When the air pressure increases, air temperature is closer to dry adiabatic because the warming effect of a given condensation heat on the upper atmosphere is smaller with increasing background air mass. (5) Air pressure also influences the strength of surface wind stresses and thereby wind-driven oceanic circulation (Yang and Dai, 2015; Duhaut and Straub, 2006). In global circulation models, the wind stresses are always parameterized as $\tau_{wind} = \rho_{air}C_D(U_s - U_o)^2$, where $\rho_{air}$ is the density of the air, $C_D$ is the wind-drag coefficient, $U_s$ is the wind speed at a certain height above the sea surface, and $U_o$ is the speed of ocean currents. A thinner (thicker) atmosphere results in weaker (stronger) wind stresses and smaller (larger) meridional oceanic heat transport, for a given surface wind speed.

In this study, we use a 1-D radiative-transfer model and a 3-D global atmospheric circulation model to examine the climatic influences of varying background atmosphere pressure under a dimmer sunlight. Our aim is to figure out how changes in atmosphere pressure affect the climate during the early period of Earth. In section 2, we introduce the model and experimental designs. In sections 3 and 4, we show the results of the radiative transfer model and the global circulation model, respectively. Section 5 is the summary.





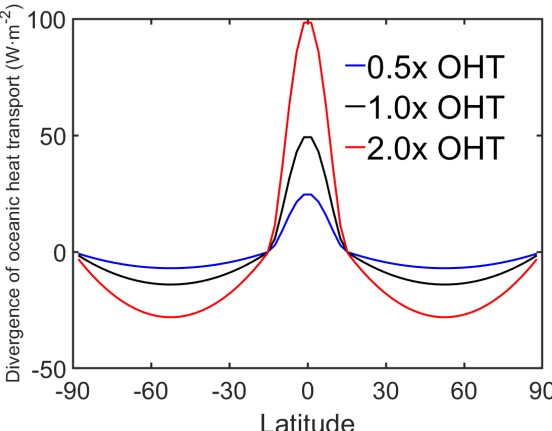

**Figure 1.** The divergence of oceanic heat transport (OHT) specified in the simulations with positive value representing a cooling effect and negative value a warming effect on the surface. Black line: the modern Earth's value but it is set to be symmetrical about the equator. Red line: twice the modern level. Blue line: half of the modern level.

## 2 Model descriptions and experimental designs

### 2.1 1-D radiative-transfer model

Firstly, we use the 1D radiative-transfer model, Climate Modelling Toolkit (CliMT, Monteiro and Caballero (2016)) to examine the radiative effect of varying surface air pressure. The main radiative effect can be divided into two parts. On one
5 hand, a thicker atmosphere leads to more Rayleigh scattering in shortwave radiation, which would be a negative forcing for surface temperature. On the other hand, a thicker atmosphere increases the absorption of greenhouse gases due to pressure broadening, leading to a positive forcing. To test the influences of these two parts under different situations, we run four groups of experiments. In the groups A and B, we test a moist atmosphere and the air temperature decreases from the surface to the top of the model following moist adiabatic (i.e., relative humidity is 100%) until the temperature reaches a minimum of 200
10 K. Above this level, the atmosphere is set to be isothermal with a temperature of 200 K and has a constant specific humidity. In the groups C and D, we test a dry atmosphere with no water vapor. The temperature profile is similar to the cases of A and B but replaced with dry adiabatic below the layer of 200 K. In the groups of A and C, the partial pressure of $CO_2$ is 33.4 Pa (close to the value of the present atmosphere), and in B and D, it is 10,000 Pa, i.e., 0.1 bar. In each group, we test a series of different surface air pressures: 0.25, 0.5, 1.0, 1.5, 2.0, 2.5, 3.0, and 3.5 bar. The model has 300 vertical levels. Through these
15 four groups of experiments, we can know the effect of air pressure on the radiative transfer of $CO_2$ and $H_2O$.

### 2.2 3-D atmospheric general circulation model

Secondly, we employed a general circulation model (GCM) to examine the influences of atmosphere background pressure and oceanic heat transport (OHT). The Community Atmosphere Model (CAM3) developed at the National Center for Atmo-



**Table 1.** 3D general circulation model settings and the simulated global-mean surface temperature. In these experiments, the effect of doubling or halving the surface pressure on the global-mean surface temperature is between 10 and 84 K, depending on the strengths of ice albedo and water vapor feedbacks.

| Oceanic heat transport | $CO_2$ partial pressure | Surface air pressure | Global-mean surface temperature |
|---|---|---|---|
| 0.5×OHT | 0.04 bar | 0.5 bar | 209 K |
|  |  | 1.0 bar | 287 K |
|  |  | 2.0 bar | 301 K |
|  |  | 4.0 bar | 318 K |
| 1.0×OHT | 0.04 bar | 0.5 bar | 211 K |
|  |  | 1.0 bar | 295 K |
|  |  | 2.0 bar | 305 K |
|  |  | 4.0 bar | 321 K |
| 2.0×OHT | 0.04 bar | 0.5 bar | 287 K |
|  |  | 1.0 bar | 300 K |
|  |  | 2.0 bar | 310 K |
|  |  | 4.0 bar | 326 K |
| 0.5×OHT | 0.06 bar | 0.5 bar | 212 K |
|  | 0.08 bar |  | 275 K |
|  | 0.10 bar |  | 284 K |
|  | 0.12 bar |  | 289 K |

spheric Research (NCAR) (Collins et al., 2004) is a widely used model to study climates of Earth's present, past, and future. The model is able to well simulate the present climate (Hurrell et al., 2006; Hack et al., 2006). CAM3 has also been used to successfully study past climates, such as the Eocene epoch (Huber and Caballero, 2011). A slab ocean module is coupled to the GCM. It allows for a fully-interactive treatment of surface energy exchange processes (Collins et al., 2004). The slab ocean

5  module represents the mixing layer of ocean and calculates the sea surface temperature, sea ice coverage and ice thickness based on surface energy balance. A thermodynamic sea ice model is coupled to the slab ocean (Collins et al., 2006). The sea ice model is based on Briegleb et al. (2004) and is used to calculate ice fraction and ice thickness. For the visible band (<0.7 $\mu$m), the snow albedo is 0.91 and ice albedo is 0.68 if surface temperature is below $-1\,°C$. For the near infrared band (>0.7 $\mu$m), it is 0.63 for snow and 0.30 for sea ice. Between $-1\,°C$ and $0\,°C$, the surface albedo decreases linearly with temperature,

10  and the albedo at $0\,°C$ is assumed to be 0.425 for sea ice and 0.656 for snow. The albedo of open ocean is varied from 0.05 to 0.1 for different solar zenith angles and is uniform for all wavelengths (Collins et al., 2004; Yang et al., 2012).

In our study, we set an aqua planet without lands. The solar constant is set to be 1,024 $W\,m^{-2}$, which is 75% of the present level. The surface pressure is varied from 0.5 to 4.0 bar in different experiments (Table 1). Bulk compositions of the atmosphere are $N_2$, $CO_2$, and $CH_4$, and Oxygen and ozone are removed. The $CH_4$ concentration is taken to be 100 Pa

15  (i.e., ≈1,385 times of the pre-industrial level). The partial pressure of $N_2O$ is 0.27 Pa. The total surface pressure is $p_{total} =$





$pN_2 + pCO_2 + pCH_4 + pN_2O + pH_2O$. By default, atmospheric $CO_2$ partial pressure is set to be 0.04 bar and the meridional OHT is set to be close to the model value (Fig. 1). Four different air pressures (0.5, 1.0, 2.0, and 4.0 bar) and three different OHT levels (0.5, 1.0, and 2.0 of the present-day level) have been tested. In our simulations, there is no land and the orbital obliquity is set to be zero, thus we set the OHT symmetrically between the northern and southern hemispheres.

Three types of cloud are considered in the model: low-level marine stratus cloud, convective cloud, and layered cloud (Collins et al., 2006). Cloud fraction of marine stratocumulus depends on the potential temperature difference between the surface and the level of 700 hPa (Klein and Hartmann, 1993). As we change the atmospheric pressure in this study, we change the critical layer from 700 hPa to 70% of surface air pressure, so that the parameterization for stratus cloud fraction has been replaced with $C_{st} = min\{1., max[0., (\theta_{0.7} - \theta_s) \times 0.057 - 0.5573]\}$, where $\theta_{0.7}$ and $\theta_s$ are the potential temperatures at the

layer where the air pressure is 70% of the surface pressure and at the surface, respectively. For example, if the surface air pressure is 0.5 bar, the parameterization will use the surface and 350-hPa potential temperatures to calculate the low-level marine stratus cloud fraction. For layered clouds, we did similar adjustments. For convective clouds, the parameterization of cloud fraction does not depend on air pressure.

## 3   Results of the 1D radiative-transfer model

The net radiative effect of increasing surface air pressure is positive when the $CO_2$ partial pressure is 33.4 Pa (same as the present level) and the atmospheric relative humidity is set to 100% (Fig. 2a). The longwave radiative effect increases with surface air pressure because of pressure broadening on the absorption lines of greenhouse gases ($CO_2$ and $H_2O$). Without pressure broadening, the natural width of gas absorption lines is quite narrow (Goldblatt, 2016; Goldblatt et al., 2009). The pressure broadening effect becomes stronger as $CO_2$ partial pressure is increased from 33.4 Pa to 0.1 bar (Fig. 2b). The

shortwave Rayleigh scattering effect increases with surface air pressure, cooling the surface. The trend of the shortwave cooling effect has no obvious difference when the $CO_2$ partial pressure is 33.4 Pa or 0.1 bar. In this calculation, the effect of pressure broadening overcomes the effect of Rayleigh scattering, so that the net radiative effect is positive. Based on these results, we know that if the Archean has a thinner atmosphere, the radiative effect would be a cooling.

For the strength of the pressure broadening effect, the concentrations of greenhouse gases are important, besides of the level

of air pressure. The warming effect due to pressure broadening weakens when water vapor is removed in the radiative-transfer calculations while the effect of Rayleigh scattering has no significant change (Fig. 3 versus Fig. 2). As the $CO_2$ partial pressure is 33.4 Pa, the effect of Rayleigh scattering dominates over pressure broadening (Fig. 3a). As the $CO_2$ partial pressure is 0.1 bar, the warming of pressure broadening and the cooling effect of Rayleigh scattering nearly cancels (Fig. 3b).





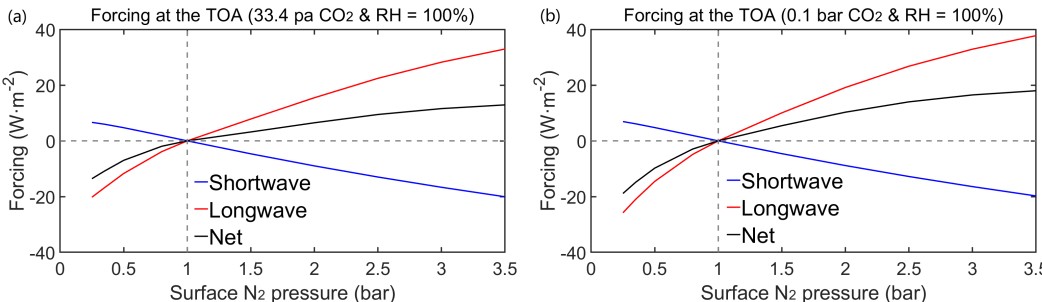

**Figure 2.** The radiative effects of varying surface air pressure in a saturated atmosphere, calculated using the 1D radiative transfer model. (a) $CO_2$ partial pressure is 33.4 Pa (close to the present level) and the solar constant is 75% of the present level. (b) same as (a) but the partial pressure of $CO_2$ is 0.1 bar. The relative humidity is set to 100% everywhere. The surface temperature is 300 K and the temperature profile follows moist adiabatic.

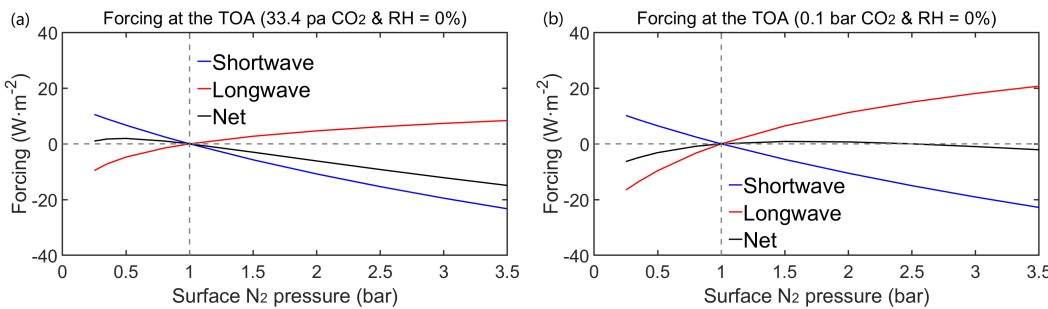

**Figure 3.** Same as Fig. 2 but for a dry atmosphere. The relative humidity is zero everywhere. The surface temperature is 300 K and the temperature profile follows dry adiabatic.

## 4 Results of the 3D GCM

### 4.1 The effect of varying surface air pressure

As shown in Fig. 4a, the surface temperature increases with increasing surface pressure. When the solar constant is set to 75% of the present level, oceanic heat transport is fixed to the modern level of Earth, surface air pressure is 1 bar and $CO_2$ partial pressure is set to 0.04 bar (about 145 times the pre-industrial level), the equator-to-pole surface temperature difference is 42 K and the global-mean surface temperature is 294 K, somewhat warmer than that of modern Earth. When the surface air pressure is decreased from 1 to 0.5 bar, the planet enters into a snowball state (blue line in Fig. 5a) due to the weakening of atmospheric greenhouse effect as addressed in the section 3 and to the strong effect of ice albedo feedback. The global-mean surface albedo is 0.71 in the 0.5 bar case whereas it is only 0.08 in the 1.0 bar case. When the surface air pressure is increased from 1.0 to 2.0 bar, the global-mean surface temperature increases by 10 K, and all the sea ice melts (green line in Fig. 5a). When the surface air pressure is further increased from 2.0 to 4.0 bar, the global-mean surface temperature increases by 17 K.



Besides of the ice albedo feedback, other processes could also influence the response of surface temperature when the surface air pressure is changed. Water vapor feedback, a positive feedback, acts to amplify the surface warming when the surface pressure is increased whereas it acts to amplify the surface cooling when the surface pressure is decreased (Fig. 6). The global-mean vertically integrated water vapor mass is 0.1, 36, 61, and 173 $\mathrm{kg\,m^{-2}}$ in the experiments of 0.5, 1.0, 2.0, and 4.0

bar, respectively.

Cloud feedback is also positive in these experiments. The global-mean net (shortwave plus longwave) cloud radiative effect at the top of the atmosphere is -9.0, -4.1, and 2.4 $\mathrm{W\,m^{-2}}$ in the experiments of 1.0, 2.0, and 4.0 bar, respectively (Fig. 7a). There are two main characteristics in the response of the clouds as increasing the air pressure. Firstly, the fraction of low-level clouds decreases whereas the fraction of high-level clouds increases, meaning that there is an upshift of the cloud system (Fig. 8a).

This is consistent with the response of the Hadley cells that reach higher levels as the surface pressures increase (Fig. 11). Secondly, the high-concentration regions of cloud water path exhibits a significant equatorward shift as increasing the surface pressure (Fig. 9a). This equatorward shift is related to the response of mid-latitude jets that is associated with the change of vertical stratification; we plan to address this in a separate paper, in which we will focus on atmospheric dynamics.

In the case of 0.5 bar, the shortwave cloud radiative effect is close to zero. This is because all of the surface is covered by ice

and snow and thereby the surface albedo is close to cloud albedo (Pierrehumbert, 2005). So that, the net cloud radiative effect in this case is dominated by the longwave cloud radiative effect, which is positive. In global mean, the net cloud radiative effect is +8.0 $\mathrm{W\,m^{-2}}$. Moreover, the cloud water path is very low in the snowball state but the cloud fraction is the highest among the four experiments. The latter is due to that relative humidity in the extremely cold snowball state is higher than that in other cases.

Figure 10 shows the meridional atmospheric energy transport, which is calculated using $2\pi R cos\theta \int_0^{Ps}(c_p T + LQ + gZ)V\frac{dp}{g}$, where $R$ is the planetary radius, $g$ is the surface gravity, $T$ is the air temperature, $Q$ is the specific humidity, $Z$ is the geopotential height, $c_p$ is the specific heat capacity, and $L$ is the latent heat of fusion. This figure shows that the atmospheric energy transport nearly doesn't change when the surface air pressure is varied. For given atmospheric meridional velocities, the meridional atmospheric energy transport should increase as increasing the air mass. However, the meridional velocities decrease

when the surface air pressure is increased (Fig. 11b, c, & d). As a result, the meridional atmospheric energy transport doesn't change obviously. An exception is the snowball case of 0.5 bar (Fig. 10a), in which the atmospheric energy transport is the lowest. This is due to the higher surface and planetary albedos and consequently less energy is required to be transported from the low latitudes to the high latitudes.

Besides of the warming of the global surface when the air pressure is increased, another obvious feature of the surface

response is that the increases of surface temperature at the polar regions are higher than that at the low latitudes (Fig. 4a). The surface temperature at the equator rises by 10 K as the surface pressure is increased from 1.0 to 2.0 bar, while the polar surface temperature rises by 23 K. The main reason is the ice albedo feedback. When the surface air pressure is 1.0 bar, the high latitudes are covered by sea ice and snow (Fig. 5a), which reflect shortwave radiation effectively. In the 2.0 bar case, all the surface ice melts, absorbing more shortwave radiation at the sea surface. Moreover, cloud feedback also acts to reduce the

equator-to-pole temperature difference, especially in the cases of 2.0 and 4.0 bar (Figs. 8 and 9). As the surface pressure is





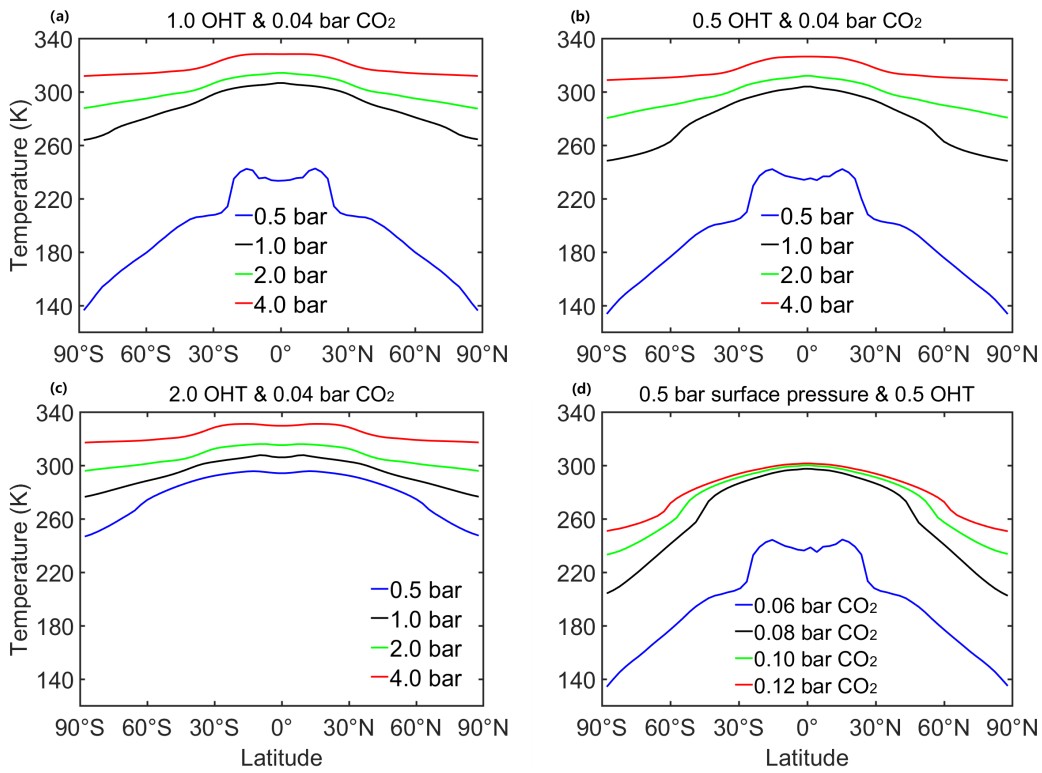

**Figure 4.** Zonal-mean surface air temperatures in all of the experiments. (a): Varying surface air pressure (0.5, 1.0, 2.0, and 4.0 bar) under fixed oceanic heat transport (OHT) (equal to the modern level, or 1.0 OHT) and fixed $CO_2$ partial pressure (0.04 bar). (b): Same as (a), but for 0.5 times the modern OHT. (c): Same as (a), but for twice the modern OHT. (d): Varying $CO_2$ partial pressure (0.06, 0.08, 0.1, and 0.12 bar) under fixed OHT (half of the modern level) and fixed surface air pressure (0.5 bar). In all these cases, the solar constant is 1,024 W m$^{-2}$, $CH_4$ partial pressure is 1 mbar, and $N_2O$ partial pressure is 0.27 $\mu$bar.

increased from 2.0 to 4.0 bar, the low-level clouds reduce in both cloud fraction and cloud water path and the high-level clouds exhibit a significant upper lift at middle and high latitudes of both hemispheres. Both these two responses act to warm the surface through reflecting less shortwave radiation from the Sun and through trapping more infrared radiation from the surface.

Although the surface temperature increases with air pressure, the precipitation rate does not exhibit the same trend. As shown in Fig. 12a, b & c, in general, the precipitation becomes weaker in the tropics but becomes stronger in the mid-latitudes as increasing the air pressure, under given oceanic heat transport and greenhouse gas concentrations. This is due to that a stronger Rayleigh scattering with increasing surface pressure leads to more shortwave reflection and less shortwave radiation reaching the surface, so that less evaporation and convection can be driven in the tropics. This mechanism was first proposed in Poulsen et al. (2015). The increasing trend of precipitation in the mid-latitudes is due to the increases of surface and air temperatures and more water vapor can be maintained in the atmosphere and supplied to rainfall.





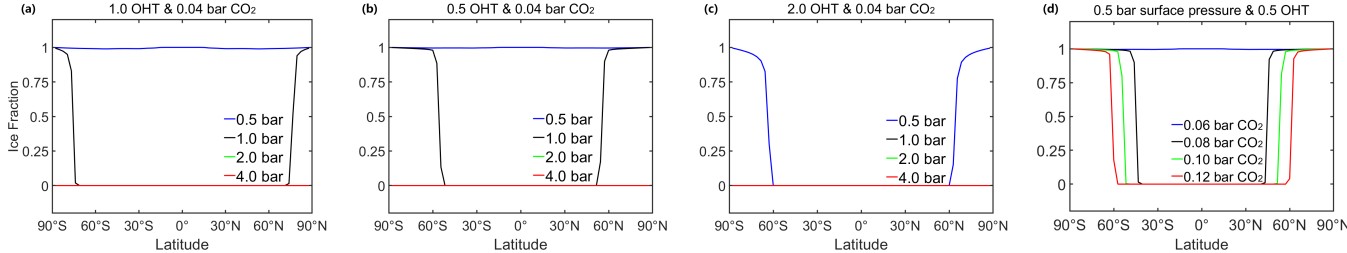

**Figure 5.** Same as Fig. 4 but for sea ice fraction. Note that the green and red lines coincide in panels (a) and (b), and the black, green, and red lines coincide in panel (c). The surface is completely ice-free in the cases of 2.0 and 4.0 bar in (a), of 2.0 and 4.0 bar in (b), and of 1.0, 2.0, and 4.0 bar in (c).

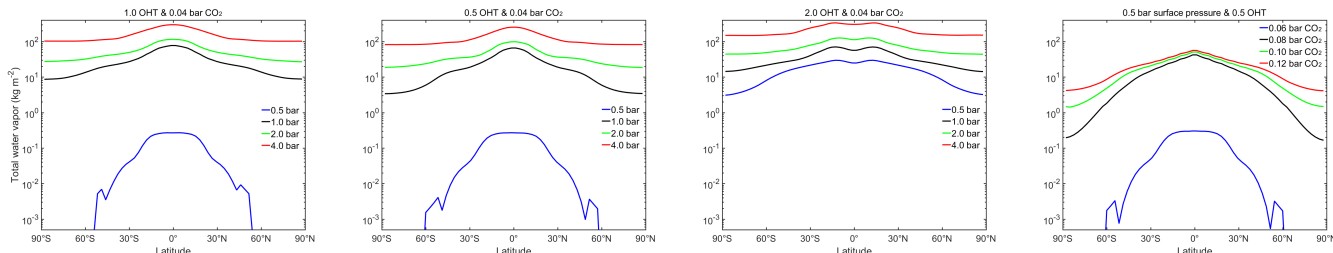

**Figure 6.** Same as Fig. 4 but for vertically-integrated zonal-mean water vapor in the atmosphere (kg m$^{-2}$).

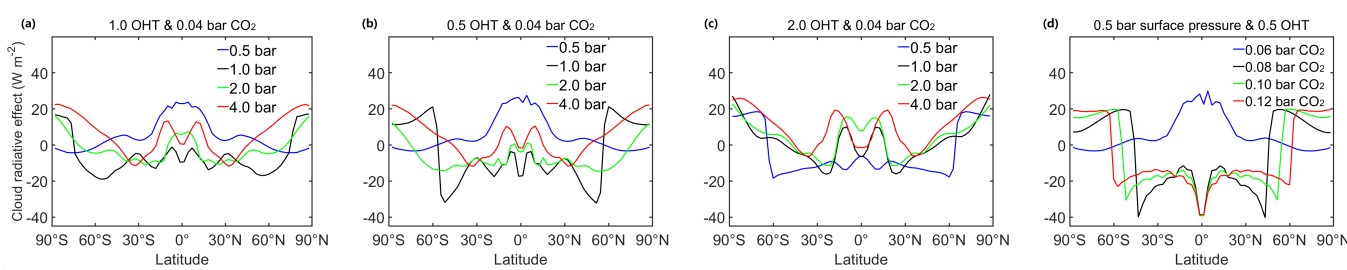

**Figure 7.** Same as Fig. 4 but for net (shortwave plus longwave) cloud radiative effect at the top of the atmosphere. In (a), the global-mean values are 8.0, -8.9, -4.1, and 2.4 W m$^{-2}$ for the blue, black, green, and red lines, respectively. In (b), the corresponding values are 8.5, -9.5, -8.6, and 0.6 W m$^{-2}$. In (c), the corresponding values are -9.1, -2.3, 2.0, and 7.0 W m$^{-2}$. In (d), the corresponding values are 8.9, -11.2, -13.0, and -14.1 W m$^{-2}$.





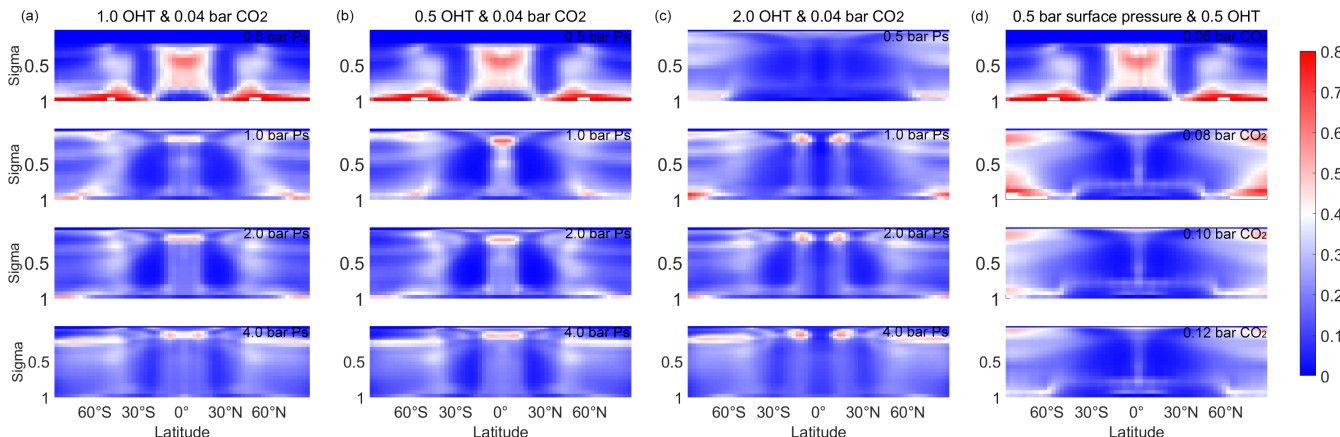

**Figure 8.** Cloud fraction (0-1) as a function of latitude and sigma coordinate ($\eta = \frac{p}{p_s}$). (a) Varying surface pressure under 1.0 OHT and 0.04 bar $CO_2$; (b) varying surface pressure under 0.5 OHT and 0.04 bar $CO_2$; (c) varying surface pressure under 2.0 OHT and 0.04 bar $CO_2$; and (d) varying $CO_2$ concentration under 0.5 OHT and 0.5-bar surface pressure.

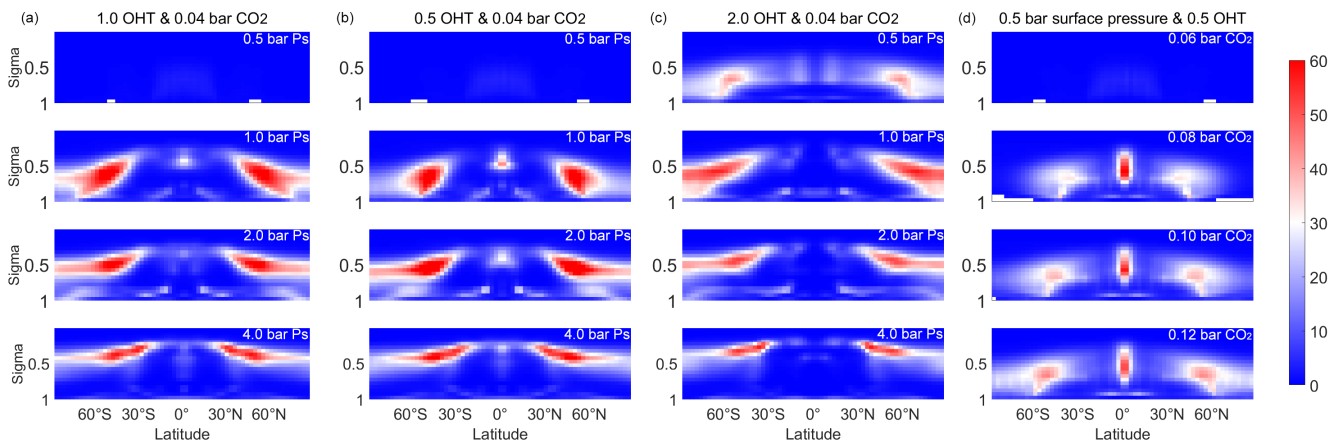

**Figure 9.** Same as Fig. 8 but for cloud water path (g m$^{-2}$).

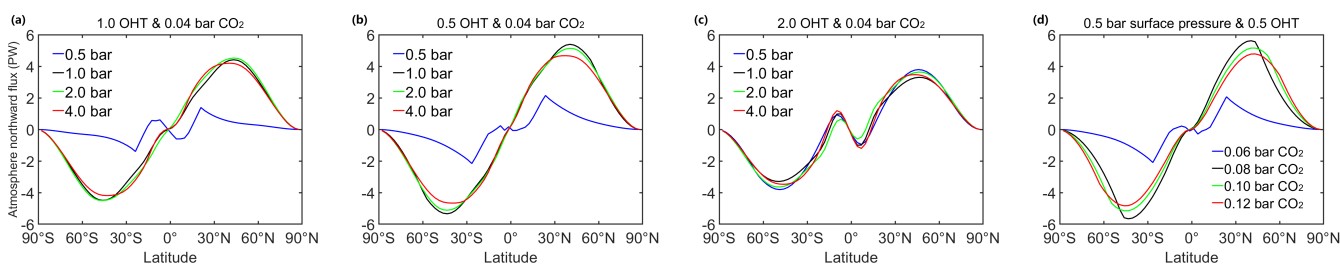

**Figure 10.** Same as Fig. 4 but for northward atmospheric energy transport (PW, 1 PW = $10^{15}$ W). The 0.5-bar surface pressure case in (a), the 0.5-bar surface pressure case in (b), and the 0.06-bar $CO_2$ case in (d) are in snowball state, so that the energy transports are small.



**Figure 11.** Same as Fig. 8 but for atmospheric mass streamfunction (color shading, in units of $10^{11}$ kg s$^{-1}$) and zonal-mean meridional winds (contour lines, with an interval of 0.5 m s$^{-1}$ and with positive (negative) values representing northward (southward). (a-d) Varying surface pressure under 1.0 OHT and 0.04 bar $CO_2$; (e-h) varying surface pressure under 0.5 OHT and 0.04 bar $CO_2$; (i-l) varying surface pressure under 2.0 OHT and 0.04 bar $CO_2$; and (m-p) varying $CO_2$ concentration under 0.5 OHT and 0.5-bar surface pressure.



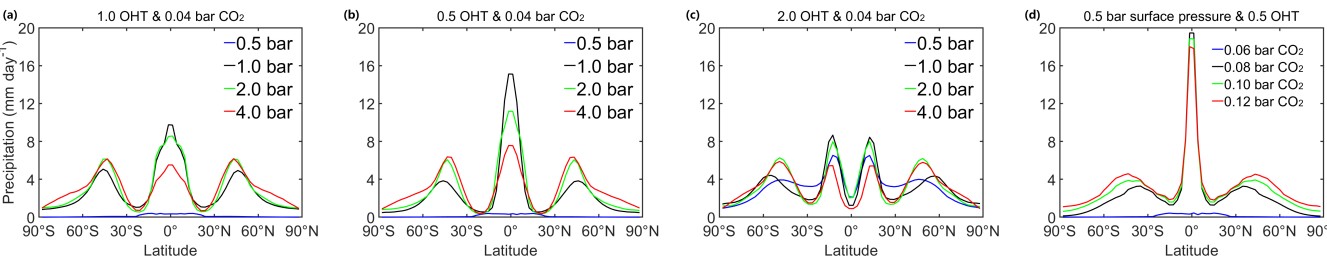

**Figure 12.** Same as Fig. 4 but for zonal-mean precipitation (mm per day).

## 4.2 The effect of varying oceanic heat transport

Oceanic heat transport (OHT) can also influence the surface temperature. In general, if the OHT is stronger, the surface becomes warmer. This is due to the fact that oceanic heat transport acts to melt the ice and snow in the high latitudes; water vapor feedback acts to further amplify the surface warming (Herweijer et al., 2005; Rose and Ferreira, 2012). For instance,

the surface ice coverage is 0.17, 0.03, and 0 (Fig. 5a, b, & c) and the global-mean surface temperature is 287, 295, and 300 K (Fig. 4a, b, & c) in the cases of 0.5, 1.0, and 2.0 times the present-day OHT under a surface pressure of 1 bar. Another example is that when the surface pressure is 0.5 bar, OHT is able to avoid the planet falling into a snowball state if it is increased from 0.5 or 1.0 to 2.0 times the present-day level.

The warming effect of increasing OHT is weaker when the high latitudes are ice-free (see the green and red lines in Fig. 4a,

b, & c). In all the experiments, the changes of surface temperature in the tropics are weaker than those at high latitudes. This result is due to the combined response of atmospheric energy transport and clouds. Atmospheric energy transport tends to compensate the change of OHT in spite of not 100% (Fig. 10a, b, & c). The atmospheric energy transport decreases as increasing OHT, same as that found in previous studies of Winton (2003), Vallis and Farneti (2009), and Barreiro et al. (2011). For the cloud feedback, the Hadley cells become weaker when OHT is increased, due to that OHT weakens the meridional

temperature gradient; as a result, both cloud fraction and cloud water path decrease in the deep tropics (Fig. 8a, b, & c and Fig. 9a, b, & c), which allows more shortwave radiation to reach the tropical surface and warms the surface, same as that found in Koll and Abbot (2013).

Interesting, in the snowball cases (blue lines in Fig. 4a & b), the surface temperature around 20°S and 20°N are higher than that in the deep tropics. This is due to the fact that the ice is covered by snow in the deep tropics and in the poleward

regions of 30°S and 30°N, but the ice is snow-free in the subtropics where sublimation is faster than snowfall. As a result, the direction of the atmospheric energy transport in the deep tropics is equatorward rather than poleward although the magnitude is small (Fig. 10a & b). Moreover, the atmospheric energy transport is also equatorward in the cases of doubling OHT as shown in Fig. 10c. This is due to that this specified value of OHT is so strong that the surface temperature in the deep tropics is somewhat lower than that in the subtropics (Fig. 4c). This result is unrealistic and implies that fully coupled atmosphere-ocean

circulation models are required in this problem.



### 4.3 How much $CO_2$ is required to maintain a temperate early Earth?

In this section, we discuss how much $CO_2$ is required to maintain a temperate climate during the early Earth. Here, we define a temperate climate for which the global-mean surface temperature is roughly within a 10-K range of the modern value, i.e., $288 \pm 10$ K. The results of this study and other GCM studies of Wolf and Toon (2013), Wolf and Toon (2014), Charnay et al.
(2013), and Le Hir et al. (2014) are summarized in Table 2. Although these GCM simulations employed different oceanic heat transports (zero or a specified heat flux), different continental configurations (aqua-planet, idealized continents, or present-day continents), different obliquities (23.4° or 0°), different rotation period (24 or 18 hours per day), we could still find some commonness. First, under a surface pressure of 1 bar, a $CO_2$ partial pressure of about 0.01 bar plus a $CH_4$ partial pressure of about 1 mbar or a $CO_2$ partial pressure of about 0.1 bar plus near-zero $CH_4$ is required to maintain a temperate climate for early
Earth during which the solar constant was 75% or 80% of the present-day level. Second, increasing air pressure acts to warm the planet whereas decreasing air pressure acts to cool the planet. Third, if the meridional OHT is stronger (weaker) than the modern value, less (more) greenhouse gases are required to sustain a temperate surface. Fourth, it seems the uncertainties in continental configuration, obliquity (as long as it does not change much), and rotation period for the early Earth do not strongly influence the global-mean climate although these factors do influence local and/or seasonal climate.

In our simulation of 1.0 bar surface pressure and 1.0×OHT, with 0.04 bar $CO_2$ and 1 mbar $CH_4$, the global-mean surface temperature reaches 295 K. This result is close to Wolf and Toon (2014), which obtains a global-mean surface temperature of 288 K with 0.032 bar $CO_2$ and 0.1 mbar $CH_4$ under 1.0 bar $N_2$ and 75% solar constant. Other studies give a combination of 0.01 bar $CO_2$ and 2 mbar $CH_4$ to reach a temperate climate under 75% solar constant (Charnay et al., 2013). In Le Hir et al. (2014), the surface temperature reaches 289 K with 0.056 bar of $CO_2$ and 1.7 $\mu$bar of $CH_4$ under 77% solar constant. When the surface pressure is 0.5 bar but the oceanic heat transport is twice the modern level, a combination of 0.04 bar $CO_2$ and 1 mbar $CH_4$ is enough to keep a temperate climate.

### 5 Summary

Here, we employed CAM3 to study the effects of varying surface air pressure, oceanic heat transport, and greenhouse gas concentrations on the Archean climate when the Sun was 25% dimmer than that in the present day. We confirm that a thicker (thinner) atmosphere leads to a warmer (cooler) climate. Oceanic heat transport also plays a significant role in the climate. A stronger oceanic heat transport can prevent the planet from a fully ice-covered snowball case. Under a surface pressure of 1.0 bar, a combination of about 0.01 bar $CO_2$ and 1 mbar $CH_4$ is required for early Earth to retain a climate similar to modern Earth. The effect of halfing or doubling the surface pressure on the global-mean surface temperature is about 10 K or even as large as 17–84 K when the ice albedo feedback or water vapor feedback is strong (Table 1). If the Archean atmosphere is thinner and the oceanic heat transport is weaker, a combination of 0.1 bar (or more) $CO_2$ and 1 mbar $CH_4$ is required to maintain a temperate climate. Future work should explore the magnitude of oceanic heat transport under different surface pressures using coupled atmosphere-ocean models and the feedback of clouds under different surface pressures using cloud-resolving models.





**Table 2.** Results of different AGCM experiments for early Earth

| References | Solar | $P_S$ [bar] | OHT | $CO_2$ [bar] | $CH_4$ [bar] | Continent | Obliquity [°] | $P_{rot}$ [hr] | $T_S$ [K] |
|---|---|---|---|---|---|---|---|---|---|
| Charnay et al. (2013) | 75% | 1.0 | simplified ocean | 0.1 | $2\times10^{-3}$ | aqua-planet | 23.4 | 24 | 292 |
| | 80% | 1.0 | simplified ocean | 0.01 | $2\times10^{-3}$ | aqua-planet | 23.4 | 24 | 287 |
| | 80% | 1.0 | simplified ocean | 0.01 | $2\times10^{-3}$ | present-day | 23.4 | 24 | 284 |
| | 80% | 1.0 | simplified ocean | 0.01 | $2\times10^{-3}$ | supercontinent | 23.4 | 24 | 285 |
| Le Hir et al. (2014) | 77% | 1.0 | diffusive ocean | 0.056 | $1.7\times10^{-6}$ | Idealized | 23.4 | 24 | 289 |
| | 77% | 1.0 | diffusive ocean | 0.112 | $1.7\times10^{-6}$ | Idealized | 23.4 | 24 | 294 |
| Wolf and Toon (2013) | 80% | 1.0 | present-day | 0.015 | $1\times10^{-3}$ | present-day | 23.4 | 24 | 285 |
| | 80% | 1.0 | present-day | 0.02 | 0 | present-day | 23.4 | 24 | 296 |
| Wolf and Toon (2014) | 75% | 1.0 | mixed-layer ocean | 0.032 | $1\times10^{-4}$ | present-day | 23.4 | 18 | 288 |
| | 75% | 1.65 | mixed-layer ocean | 0.015 | $1\times10^{-4}$ | present-day | 23.4 | 18 | 288 |
| | 80% | 0.86 | mixed-layer ocean | 0.06 | 0 | present-day | 23.4 | 18 | 286 |
| | 80% | 1.0 | mixed-layer ocean | 0.06 | 0 | present-day | 23.4 | 18 | 288 |
| | 80% | 1.26 | mixed-layer ocean | 0.06 | 0 | present-day | 23.4 | 18 | 290 |
| | 80% | 1.65 | mixed-layer ocean | 0.06 | 0 | present-day | 23.4 | 18 | 293 |
| | 80% | 2.44 | mixed-layer ocean | 0.06 | 0 | present-day | 23.4 | 18 | 298 |
| This work | 75% | 1.0 | 0.5×OHT | 0.04 | $1\times10^{-3}$ | aqua-planet | 0 | 24 | 287 |
| | 75% | 1.0 | 1.0×OHT | 0.04 | $1\times10^{-3}$ | aqua-planet | 0 | 24 | 295 |
| | 75% | 0.5 | 2.0×OHT | 0.04 | $1\times10^{-3}$ | aqua-planet | 0 | 24 | 287 |
| | 75% | 0.5 | 0.5×OHT | 0.10 | $1\times10^{-3}$ | aqua-planet | 0 | 24 | 284 |
| | 75% | 0.5 | 0.5×OHT | 0.12 | $1\times10^{-3}$ | aqua-planet | 0 | 24 | 289 |

*Code and data availability.* The model CAM3 can be downloaded from http://www.cesm.ucar.edu/models/atm-cam/. Requirements for the changes of model source codes can send an email to XJY: jybear@pku.edu.cn.

*Author contributions.* JY lead the project. XJY performed the simulations, did the analyses, and wrote the first version of the manuscript. JY designed the experiments and improved the manuscript.

5 *Competing interests.* The authors declare no competing interests.



*Acknowledgements.* We are grateful to Yonggang Liu and Yongyun Hu for their helpful discussions and suggestions. J.Y. is supported by the National Science Foundation of China (NSFC) grant 41675071.



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
