# Peer review of "Examining the role of varying surface pressure in the climate of early Earth"

_Climate of the Past, 2020_

## Short Comment (SC1) · 19 Jul 2020

Xiong and Yang presented interesting modeling results on the role of varying surface pressure in changing Earth's temperature, which have implications on the "faint young Sun paradox". The authors' calculations using 1-D radiative-transfer model and 3-D general circulation model (GCM) suggest that increasing surface pressure warms Earth's surface due to a stronger pressure broadening effects associated with greenhouse gases. For example, their GCM simulations show a climate sensitivity of ∼10 K per doubling or halving surface pressure. Role of ocean heat transport is also discussed.

The manuscript is in general well-written and easy to follow. The research topic is inter-

esting and adds to the discussion on mechanisms for the evolution of Earth's climate. However, this manuscript in its current form is very descriptive and lacks in-depth analysis to clarify contributions from different feedback processes and to better support the authors' interpretation of results. For occasions, a more detailed description of model and experimental setup is needed. These issues should be resolved before the manuscript can be published in Climate of the Past. Please see my detailed comments below.

Major comments: 1. The radiation calculation. The authors fail to provide necessary information for readers to assess the performance of their radiation schemes in the 1-D model and the GCM. How complex is the 1-D radiative transfer model? Have the authors validated the solution against comprehensive line-by-line calculations? Related information is also required for assessing the highly parameterized and tuned radiation schemes in GCM, especially when the authors are using them well away from the climate conditions for which they were tuned. Another related question, is the radiation scheme the same between the 1-D model and the GCM?

2. On multiple occasions, the authors attribute the temperature changes in their simulations to climate feedbacks, such as ice albedo, water vapor, and cloud feedback, but fail to substantiate their claims in a quantitative manner. I understand that a complete feedback quantification for multiple GCM simulations demands large amounts of resources, but there are cheaper solutions, e.g. the approximated partial radiative perturbation (APRP) method (Taylor et al., 2007). Although not providing a complete quantification, APRP can quantify the shortwave feedbacks really well, which, in my opinion, will offer important insights on temperature responses in the authors' simulations.

3. Role of ocean heat transport (OHT). Based on the description of experimental design, it is unclear how the mixed layer depth is prescribed, a constant everywhere, or a present-day spatial distribution? The authors have acknowledged the limitation of their approach using a slab ocean model with prescribed OHT (e.g. last paragraph on Page

13), but more discussion should be added on this. First, changing OHT while fixing the mixed layer depth is not a physically consistent approach. Ocean circulation and heat transport are usually accompanied with distinct ocean structures including mixed layer conditions. For example, ocean circulation and heat transport are greater in the present-day North Atlantic, so is the mixed layer depth. Second, the physical consistency between the prescribed OHT and the climate state should be better discussed. Is an OHT of 0.5–1.0 times the present-day value possible under a cold climate with a global mean temperature of $\sim$210K? Similarly, are the OHT values realistic in a warm climate of $\sim$326K? How does a snow/ice cap impacts OHT? Is it possible that warming and freshening under a warm climate increases the ocean stratification and decreases the high-latitude OHT, making some of the authors calculations unrealistic?

Minor comments: 1. Page 1, Line 23: a low-$\delta$18O sediment infers a high ocean temperature 2. Page 2, Line 1–4: Another important caveats regarding the isotopic thermometry is the assumptions on isotopic composition of seawater, i.e. a low calcium $\delta$18O may reflect a low seawater $\delta$18O. This should be added to the discussion. 3. Page 3, Line 3–28: Poulsen, Tabor, & White (2015) is worth mentioning when reviewing findings in previous studies. 4. Page 5, Line 3: I would not say the application of CCSM3-CAM3 was successful for the Eocene. Caballero and Huber (2013) and later studies clearly showed that Eocene climate in CCSM3 is too cold when the estimated Eocene CO2 is used. 5. Model description: Please add information on model resolution and integrations length. Have the slab ocean simulations reached equilibrium? 6. Page 8, Line 22–23: the atmospheric energy transport change little when the surface air pressure is varied. 7. Page 8, Line 25–26: the meridional atmospheric energy transport does not change much. 8. Page 8, Line 29: Besides the warming of the global surface.

Reference: Taylor, K. E., Crucifix, M., Braconnot, P., Hewitt, C. D., Doutriaux, C., Broccoli, A. J., ... & Webb, M. J. (2007). Estimating shortwave radiative forcing and response in climate models. Journal of Climate, 20(11), 2530-2543.

---

## Referee Comment (RC1) · Anonymous Referee #1 · 28 Sep 2020

Recommendation: Rejected

The paper is well written and presents clearly results concerning solutions to the Faint Young Sun Problem (FYSP). However I identified a fundamental issue requiring clarification. Indeed the authors use ClimT model, an Earth system modelling toolkit, and CAM3 (a General Circulation Model) for investigating extreme climate conditions without presenting diagnostics showing the validity of their radiative scheme. For instance, the collision-induced absorption is of great importance to the overall radiative budget in dense atmospheres, but its representation in climate models remains uncertain. If RRTGM (the radiative scheme implemented in ClimT) is a state of the art radiative transfer code (and used in many climate models), that not means that this component is adapted for this specific purpose mainly due to a lack of accurate experimental and

theoretical data to explore the early Earth (and especially the surface pressure). This point is not easy to solve which explains why I recommend "rejected" rather than "major revision". If the authors want to solve this issue, the methodology is described in Wolf and Toon 2013 (study also using CAM3). Consequently sections 2.1 and 2.2 should describe the general behavior of the radiative schemes AND sets of results demonstrating the validation.

In addition here is a list a suggestions to improve the manuscript

- line 4 p6. Why the obliquity is set to 0. ?

- line 3 p 5. Citations concerning the Eocene epoch are irrelevant here (to my knowledge the surface pressure is assumed held constant and the load in carbon dioxide does not overcome 1120ppmv, so very far from values used in the present study)

- p14 section 4.3. The discussion deserves more attention. As summarized by Charnay et al. 2020 (a review paper entitled "Is the FYSP for Earth Solved ?") the explanation of a temperate early Earth is not problematic anymore (as illustrated in the table 2 p15). Despite the cooling provided by the decreasing surface pressure (table 1 p5), this section does not conclude if the FYSP becomes more (or very) problematic to solve.

- p15 table 2 (Charnay et al. 2013 and Le Hir et al. 2014 both used a mixed-layer ocean (with Ekman transport for Charnay et al. so more complex than a standard mixed-layer model). please correct this point.

---

## Referee Comment (RC2) · Anonymous Referee #2 · 6 Mar 2021

Focusing on the role of atmospheric pressure, Xiong and Yang present one possible solution for the faint young sun paradox. Overall I think this contribution can move forward our understanding and I think it should eventually be published. However, I have several concerns and questions that I think should be addressed before publication (outlined below).

Major issues:

At page 2, line 21: First off, there is no geochemical proxy on atmospheric methane, so we simply don't know their upper or lower limit in the past. Second, Pavlov et al. 2001 on Archean kerogens didn't give an upper limit on methane concentration after their modelling exercise. Third, even if they did, many of the Archean kerogens are now believed to be contaminated by the oil drilling, therefore became an unreliable indicator

for CO2/CH4 ratio.

At page 4, section 2.1: A question to the authors: the 1-D radiative transfer model also has Rayleigh scattering induced changes in planetary albedo, which then linked to the outgoing solar radiation. Did the albedo from 3-D model then coupled with the albedo parameter in the 1-D model? If not, why?

At page 15, table 2: Even if the authors can ignore the Archean high obliquity hypothesis, why is the obliquity is set to zero? Some justification is needed. Also, if ocean heat transport is a major parameter that differs from previous modeling work, what are the reasons the authors had in choosing their parameter space? Please provide more justification on the benefits of the utilized model and note how it compares to other models.

At page 5, line 14-15: even if pCH4 can be set as high as 1E-3 as a modelling exercise, I wonder why the authors didn't mention the concurrent hydrogen flux (or the lack thereof), which according to Kharecha et al., 2005 Geobiology paper, is quantitatively similar to the methane concentration (on a related note, the lead author from the same research group believe the methane estimate in their Kharecha et al. 2005 paper is more reliable than their Pavlov et al. 2001 paper, on top of my major issue 1) . Since this article is mainly about the effects of pressure, neglecting a major constitute in the Archean atmosphere seems a bit odd to me. Even if hydrogen eventually escape from the atmosphere, it is still a major constituent in the Archean atmosphere if outgassing is continuous. In addition, hydrogen serves as an indirect greenhouse gas that increases the lifetime of methane through scavenging radicals like OH.

Minor issues:

At page 1, line 22-27: One fundamental aspect about seawater temperature reconstructions the author didn't mention is that the delta 18O value in seawater can change overtime. Recent analysis on iron oxides, a temperature insensitive sedimentary proxy, shows that the seawater delta 18O value can increase by 15 permil since the Archean

(Galili et al., 2019 Science).

At page 2, line 7-13: in the texts above, the authors argued from multiple lines that the Archean seawater temperature was similar or higher than the modern value. If so, why do they argue the higher pCO2 was maintained by a low surface temperature? The authors argument based on silicate weathering feedback seemingly contradict with their own propositions on surface temperature and pCO2. It may be that this section just needs to be rewritten for clarity.

At page 2, line 24-25: it might be better to reference Pavlov and Kasting 2002 Astrobiology paper for Archean pO2. That paper was the original work that provided the most commonly cited upper limit on Archean pO2. Also, 1% PAL of O2 would contradict the modeling decision of not including oxygen and ozone in their bulk atmosphere composition, which also have pressure broadening effect on CO2 and H2O.

---

## Author Comment (AC1) · 1 May 2021

Dear Editor and Referee,

Thank you very much for handling the review on our manuscript "Examining the role of varying surface pressure in the climate of early Earth" (No. cp-2020-55). Your comments have been very helpful for improving the manuscript. In the following, we present replies (in black) to your comments (in blue). Following your comments and suggestions, we have made improvements in the revised manuscript (in red face; we will submit the revised manuscript soon if applicable).

Sincerely,

Jun Yang and Junyan Xiong,

April 30, 2021

================================================================

Response to Referee #1 cp-2020-55-SC1

Xiong and Yang presented interesting modeling results on the role of varying surface pressure in changing Earth's temperature, which have implications on the "faint young Sun paradox". The authors' calculations using 1-D radiative-transfer model and 3-D general circulation model (GCM) suggest that increasing surface pressure warms Earth's surface due to a stronger pressure broadening effects associated with greenhouse gases. For example, their GCM simulations show a climate sensitivity of 10 K per doubling or halving surface pressure. Role of ocean heat transport is also discussed. The manuscript is in general well-written and easy to follow. The research topic is interesting and adds to the discussion on mechanisms for the evolution of Earth's climate.

Reply: Thanks for these encouraging comments.

However, this manuscript in its current form is very descriptive and lacks in-depth analysis to clarify contributions from different feedback processes and to better support the authors' interpretation of results. For occasions, a more detailed description of model and experimental setup is needed. These issues should be resolved before the manuscript can be published in Climate of the Past. Please see my detailed comments below.

Reply: Following the comments and suggestions of the referee and the other two referees, we have improved the analyses of different feedbacks and have added more descriptions of the experimental design, as shown in the following responses.

Major comments:

1. The radiation calculation. The authors fail to provide necessary information for readers to assess the performance of their radiation schemes in the 1-D model and the GCM. How complex is the 1-D radiative transfer model? Have the authors validated the solution against comprehensive line-by-line

calculations? Related information is also required for assessing the highly parameterized and tuned radiation schemes in GCM, especially when the authors are using them well away from the climate conditions for which they were tuned. Another related question, is the radiation scheme the same between the 1-D model and the GCM?

Reply: We use the same radiation scheme in our 1-D calculations and 3-D GCM simulations, which is the radiation scheme of CAM3 (Briegleb 1992; Collins et al. 2002, 2004). For shortwave radiation, the solar spectrum is divided into 19 discrete spectral and pseudo-spectral intervals, 7 for $O_3$, 1 for visible, 7 for $H_2O$, 3 for $CO_2$, and 1 for the near-infrared. The continuum of $H_2O$ lines is treated with the Clough, Kneizys, and Davies (CKD) model version 2.4.1 (Clough et al. 1989). The absorption coefficients of the model are based on HITRAN2000 database (Rothman et al. 2003), which is relatively old and would cause inaccuracy in the radiative transfer calculations as shown below. But, this inaccuracy is much smaller than the uncertainty in cloud parameterizations (Cess et al. 1990, 1996; Yang et al. 2016, 2019) and the setup in snow/ice albedos (Pierrehumbert et al. 2011). For longwave radiation, the radiative transfer calculations are based on the absorptivity/emissivity formulation of Ramanathan and Downey (1986). The effect of pressure broadening on the mean line width of the bands is included. Six absorbers, $H_2O$, $CO_2$, $O_3$, $CH_4$, $N_2O$, and CFCs, are included in the model. Overlaps between $CO_2$ and $H_2O$, $CH_4$ and $N_2O$, $N_2O$ and $H_2O$, and $N_2O$ and $CO_2$ have also been considered in the calculations. The two minor bands of $CO_2$ at 961 $cm^{-1}$ and 1064 $cm^{-1}$, important for high levels of $CO_2$ such as during the Archean eon, have also been included in the model (Collins et al. 2004).

In Yang et al. (2016), they compared the radiative transfer module of CAM3 with other radiative transfer models as well as two line-by-line radiative transfer models (SMART and LBLRTM). The results are shown in Figures A1 and A2 below. These comparisons showed that at low temperatures, the differences among the models are small, but at high temperatures (>320 K), the differences are relatively large. At 280 K, differences in longwave and shortwave radiation fluxes under clear-sky conditions between CAM3 and the two line-by-line radiative-transfer models are less than 10 W $m^{-2}$, and at 300 K, it is less than 15 W $m^{-2}$. The upper limit of $CO_2$ amount that CAM3 can well simulate is about 0.1 bar (Pierrehumbert 2005; Abbot et al. 2013); most of our experiments shown in the manuscript are less than this level.

In the study here, most of our simulations have surface temperatures equal to or lower than 310 K (except the 4.0 bar experiments within which the global-mean surface temperature is higher than 310 K, see in Table 1 of the manuscript), so the model CAM3 is roughly suitable for investigating the effects of varying surface pressure, although the radiative transfer module is not as accurate as other general circulation models (such as LMDG and CAM4_Wolf) and the two line-by-line radiation transfer models. In future work, we will employ the model of CAM4_Wolf or called ExoCAM.

[Figure]

Figure A1. Outgoing longwave radiation at the top of the atmosphere for different radiative transfer models. CAM3, CAM4_Wolf, LMDG, and AM2 are the radiation transfer modules used in atmospheric general circulation models; SBDART is an independent radiation transfer model; and SMART and LBLRTM are line-by-line radiation transfer models. The surface temperature is set to be 250, 273, 300, 320, 340, and 360 K. The atmosphere is assumed to Earth-like (1 bar $N_2$, variable $H_2O$, and 376 ppmv $CO_2$). The temperature structures are moist adiabatic profiles overlain by a 200 K isothermal stratosphere. The atmosphere is assumed to be saturated in water vapor (relative humidity is equal to 100%). The volume mixing ratio of water vapor in the stratosphere is set equal to its value at the tropopause. This figure is from Yang et al. (2016).

[Figure]

Figure A2. Upward shortwave flux at the top of the atmosphere (TOA) from different radiation transfer models. The experimental designs are same as those in Figure A1. The incoming stellar radiation at TOA is 340 W m$^{-2}$, and the solar spectra is used in these calculations. This figure is from Yang et al. (2016).

2. On multiple occasions, the authors attribute the temperature changes in their simulations to climate feedbacks, such as ice albedo, water vapor, and cloud feedback, but fail to substantiate their claims in a quantitative manner. I understand that a complete feedback quantification for multiple GCM simulations demands large amounts of resources, but there are cheaper solutions, e.g., the

approximated partial radiative perturbation (APRP) method (Taylor et al., 2007). Although not providing a complete quantification, APRP can quantify the shortwave feedbacks really well, which, in my opinion, will offer important insights on temperature responses in the authors' simulations.

Reply: Thank you very much for the suggestion of using the "Approximate Partial Radiative Perturbation" (APRP) method. We did the analyses and the results are shown in Table A1 below. From this table, there are several significant findings. (1) In the three coldest states (0.5 OHT, 0.5 bar surface pressure, 0.04 bar $CO_2$; 1.0 OHT, 0.5 bar, 0.04 bar $CO_2$; and 0.5 OHT, 0.5 bar, 0.06 bar $CO_2$), the largest change in the shortwave radiation, above -100 W m$^{-2}$ (comparing to the cases of 1.0 bar), is related to surface albedo. This is because these three cases enter a globally ice-covered snowball state. In other cases, the surface albedo feedback is positive, acting to warm the surface when the air pressure or $CO_2$ concentration is increased. (2) In the snowball states, the cloud albedo effect is positive, warming the surface, because of the largely decreased cloud water path and the high surface albedo. In the first three groups of experiments, the cloud albedo feedback is negative when the surface pressure is increased to be higher than 1.0 bar, acting to cool the surface. This is likely due to the fact that the cloud water path in the atmosphere increases with temperature. (3) The change of no-cloud (clear-sky) shortwave radiation flux is due to the combined effect of the changes of water vapor amount and of Rayleigh scattering of the dry air. In general, when the background air pressure is larger, the Rayleigh scattering is stronger, promoting a cooling effect on the surface. Exceptions are the three snowball experiments within which the no-cloud shortwave radiation change is negative with a large magnitude, comparing to the case of 1.0 bar. This is due to the fact that in the snowball state specific humidity of the atmosphere is low, so that shortwave absorption by water vapor decreases greatly.

Table A1: The radiative response simulated by APRP method (Taylor et al. 2007). dQα, dQcld, dQcs is the surface albedo, cloud, and no-ncloud (clear-sky) atmospheric shortwave radiative responses, respectively.

| Group | Case | $dQ_\alpha$(W m$^{-2}$) | $dQ_{cld}$(W m$^{-2}$) | $dQ_{cs}$(W m$^{-2}$) |
|---|---|---|---|---|
| | 0.5 bar - 1.0 bar | -112.3 | 21.1 | -18.1 |
| 0.5×OHT | 2.0 bar - 1.0 bar | 9.0 | -0.6 | -11.3 |
| | 4.0 bar - 1.0 bar | 7.2 | -6.6 | -23.6 |
| | 0.5 bar - 1.0 bar | -121.7 | 15.6 | -15.3 |
| 1.0×OHT | 2.0 bar - 1.0 bar | 2.9 | 3.8 | -14.3 |
| | 4.0 bar - 1.0 bar | 2.7 | -1.0 | -27.5 |
| | 0.5 bar - 1.0 bar | 0.1 | 12.9 | 1.8 |
| 2.0×OHT | 2.0 bar - 1.0 bar | 2.2 | 1.6 | -13.1 |
| | 4.0 bar - 1.0 bar | 2.1 | -5.0 | -26.3 |
| | 0.06 bar $CO_2$ - 0.08 bar $CO_2$ | -103.5 | 18.5 | -15.4 |
| different $CO_2$ | 0.10 bar $CO_2$ - 0.08 bar $CO_2$ | 8.3 | 14.9 | -0.2 |
| | 0.12 bar $CO_2$ - 0.08 bar $CO_2$ | 12.6 | 15.0 | 4.5 |

3. Role of ocean heat transport (OHT). Based on the description of experimental design, it is unclear how the mixed layer depth is prescribed, a constant everywhere, or a present-day spatial distribution? The authors have acknowledged the limitation of their approach using a slab ocean model with prescribed OHT (e.g. last paragraph on Page 13), but more discussion should be added on this. First, changing OHT while fixing the mixed layer depth is not a physically consistent approach. Ocean circulation and heat transport are usually accompanied with distinct ocean structures including mixed layer conditions. For example, ocean circulation and heat transport are greater in the present-day North Atlantic, so is the mixed layer depth. Second, the physical consistency between the prescribed OHT and the climate state should be better discussed. Is an OHT of 0.5–1.0 times the present-day value possible under a cold climate with a global mean temperature of ~210 K? Similarly, are the OHT values realistic in a warm climate of 326 K? How does a snow/ice cap impacts OHT? Is it possible that warming and freshening under a warm climate increases the ocean stratification and decreases the high-latitude OHT, making some of the authors calculations unrealistic?

Reply: Our simulations use a 50-m slab ocean and it is constant everywhere. The oceanic heat transport (OHT) is prescribed in our simulations and fixed in time. Through employed different magnitudes of OHT, we examined the effect of OHT on the surface climate. This is the first step in knowing the sensitivity of the climate system to OHT, and it is a general method used in paleoclimate studies when coupled atmosphere-ocean model experiments are computer source limited.

For the case of a global-mean temperature of 210 K, all the surface is covered by ice and snow and the planet is in a snowball state. In the snowball state, the OHT should be much less than 0.5 or 1.0 OHT, although geothermal heat-driven ocean circulation is still robust (Ashkenazy et al. 2013). However, in this study, our main concern is not the exact climate state during the snowball state but the transition phase from an ice-free state or a partly ice-covered state to a snowball state; during the transition, a 0.5-1.0 OHT is not very extraordinary, following the previous simulations of Poulsen and Jacob (2004) and Yang et al. (2012).

For the warm case of 326 K, the OHT should be lower than present, due to the reduced surface temperature gradients and weakened surface winds. So, the 1.0 or 2.0 OHT and the fixed mixed layer depth used in this study are unrealistic. But, again, our main concern is not the exact climate state after all ice/snow melts but the transition phase from a partly ice-covered state to an ice-free state. Of course, we agree that fully coupled atmosphere-ocean models are required to further investigate this problem.

Recently, we are trying to use the coupled atmosphere-ocean model CESM 1.2.2 to simulate the effect of varying surface pressure on the climate of the Archean eon. Preliminary result is shown in Figure A3. As shown in the figure, lowering the surface pressure from 1.0 bar to 0.5 bar, the planet enters a fully ice-covered snowball state; this result is similar to that found in the CAM3 experiments (see Table 1 of the manuscript). We will present the coupled simulations in a near future paper.

[Figure]

Figure A3. Time series of global-mean surface air temperature (a) and energy balance at the top of the atmosphere (TOA, (b)) in the fully coupled atmosphere-ocean experiments using the model CESM1.2.2. The surface pressure is 0.5 bar for the blue line and 1.0 bar for the red line. The stellar flux is 1024 W m$^{-2}$ (75% of the present-day value), $CO_2$ partial pressure is 0.04 bar, and $CH_4$ partial pressure is 1 mbar. The land-sea configuration of 1520 Ma is used in these two experiments because older land-sea configuration during the Archean is unknown. The planetary obliquity is 23.5°.

Minor comments:

1. Page 1, Line 23: a low-$\delta^{18}$O sediment infers a high ocean temperature.

Reply: Corrected.

2. Page 2, Line 1–4: Another important caveat regarding the isotopic thermometry is the assumptions on isotopic composition of seawater, i.e. a low calcium $\delta^{18}$O may reflect a low seawater $\delta^{18}$O. This should be added to the discussion.

Reply: Added.

3. Page 3, Line 3–28: Poulsen, Tabor, & White (2015) is worth mentioning when reviewing findings in previous studies.

Reply: We added several sentences in the section of Conclusion and Discussions to address the paper of Poulsen et al. (2015): "The study of Poulsen et al. (2015) showed that varying surface pressure has significant effects on both surface temperature and precipitation. When the surface pressure is

decreased (such as due to a lower concentration of $O_2$), Rayleigh scattering of the atmosphere decreases and more solar radiation reaches the surface; the increased surface solar radiation drives stronger convection and produces more precipitation. This is consistent with our results. But, Poulsen et al. (2015) found that the surface temperature increases with reducing surface pressure, which is opposite to the conclusion shown in our study and other previous studies such as Goldblatt et al. (2009), Wolf and Toon (2014), and Paradise et al. (2021): the surface temperature decreases with reducing surface temperature. The main contrast is from the opposite cloud feedbacks. Poulsen et al. (2015) showed that the cloud feedback associate with low-level marine stratus acts to warm the surface when the surface pressure is lowered. But, in our simulations and other previous studies, the cloud feedback acts to warm the surface when the surface pressure is increased. These contrast results also suggest that future work with limited-area or global cloud resolving models are required to examine the trend of clouds under different background air pressures." We have also cited this article in section 4.1 where we discussed the effect of varying surface pressure on precipitation.

4. Page 5, Line 3: I would not say the application of CCSM3-CAM3 was successful for the Eocene. Caballero and Huber (2013) and later studies clearly showed that Eocene climate in CCSM3 is too cold when the estimated Eocene $CO_2$ is used.

Reply: Corrected. We deleted this sentence in the revised manuscript.

5. Model description: Please add information on model resolution and integrations length. Have the slab ocean simulations reached equilibrium?

Reply: We added three sentences to more clearly describe the model and the experimental design: "The model has 26 levels in vertical and the horizontal resolution is approximately 2.8° in longitude by 2.8° in latitude. Each experiment was run for 60 Earth years. The model always reached equilibrium within 50 years, and the averages of the final 10 years were used to analyses below."

6. Page 8, Line 22–23: the atmospheric energy transport change little when the surface air pressure is varied.

Reply: Corrected.

7. Page 8, Line 25–26: the meridional atmospheric energy transport does not change much.

Reply: Corrected.

8. Page 8, Line 29: Besides the warming of the global surface.

Reply: Corrected

References:

1.  Abbot, D. S., Voigt, A., Li, D., Hir, G. L., Pierrehumbert, R. T., Branson, M., ... & B. Koll, D. D. (2013). Robust elements of Snowball Earth atmospheric circulation and oases for life. Journal of Geophysical Research: Atmospheres, 118(12), 6017-6027.

2.  Ashkenazy, Y., Gildor, H., Losch, M., Macdonald, F. A., Schrag, D. P., & Tziperman, E. (2013). Dynamics of a Snowball Earth ocean. Nature, 495(7439), 90-93.

3.  Briegleb, B. P., Delta-Eddington approximation for solar radiation in the NCAR Community Climate Model, J. Geophys. Res., 97, 7603–7612, 1992.

4.  Cess, R. D., et al. (1990). Intercomparison and interpretation of climate feedback processes in 19 atmospheric general circulation models. Journal of Geophysical Research: Atmospheres, 95(D10), 16601-16615.

5.  Cess, R. D., et al. (1996). Cloud feedback in atmospheric general circulation models: An update. Journal of Geophysical Research: Atmospheres, 101(D8), 12791-12794.

6.  Clough, S. A., Kneizys, F. X., & Davies, R. W. (1989). Line shape and the water vapor continuum. Atmospheric research, 23(3-4), 229-241.

7.  Collins, W. D., Hackney, J. K., & Edwards, D. P. (2002). An updated parameterization for infrared emission and absorption by water vapor in the National Center for Atmospheric Research Community Atmosphere Model. Journal of Geophysical Research: Atmospheres, 107(D22), ACL-17.

8.  Collins, William D., et al. "Description of the NCAR community atmosphere model (CAM 3.0)." NCAR Tech. Note NCAR/TN-464+ STR 226 (2004).

9.  Goldblatt, C., Claire, M. W., Lenton, T. M., Matthews, A. J., Watson, A. J., & Zahnle, K. J. (2009). Nitrogen-enhanced greenhouse warming on early Earth. Nature Geoscience, 2(12), 891-896.

10. Paradise, A., Fan, B. L., Menou, K., & Lee, C. (2021). Climate diversity in the solar-like habitable zone due to varying background gas pressure. Icarus, 358, 114301.

11. Pierrehumbert, R. T. (2005). Climate dynamics of a hard snowball Earth. Journal of Geophysical Research: Atmospheres, 110(D1).

12. Pierrehumbert, R. T., Abbot, D. S., Voigt, A., & Koll, D. (2011). Climate of the Neoproterozoic. Annual Review of Earth and Planetary Sciences, 39, 417-460.

13. Poulsen, C. J., & Jacob, R. L. (2004). Factors that inhibit snowball Earth simulation. Paleoceanography, 19(4).

14. Poulsen, C. J., Tabor, C., & White, J. D. (2015). Long-term climate forcing by atmospheric oxygen concentrations. *Science*, *348*(6240), 1238-1241.

15. Ramanathan, V., & Downey, P. (1986). A nonisothermal emissivity and absorptivity formulation for water vapor. Journal of Geophysical Research: Atmospheres, 91(D8), 8649-8666.

16. Rothman, L. S., et al. (2003). The HITRAN molecular spectroscopic database: edition of 2000

including updates through 2001. Journal of Quantitative Spectroscopy and Radiative Transfer, 82(1-4), 5-44.

17. Taylor, K. E., Crucifix, M., Braconnot, P., Hewitt, C. D., Doutriaux, C., Broccoli, A. J., Mrrchell, J. F. B. & Webb, M. J. (2007). Estimating shortwave radiative forcing and response in climate models. *Journal of Climate*, *20*(11), 2530-2543.

18. Wolf, E. T., & Toon, O. B. (2014). Controls on the Archean climate system investigated with a global climate model. Astrobiology, 14(3), 241-253.

19. Yang J., W. Richard Peltier, and Yongyun Hu: The initiation of modern "soft Snowball" and "hard Snowball" climates in CCSM3. Part II: climate dynamic feedbacks. *J. Climate*, 2012, 25, 2737-2754.

20. Yang, J., Leconte, J., Wolf, E. T., Goldblatt, C., Feldl, N., Merlis, T., Wang, Y., Koll, D. D. B., Ding, F., Forget, F. & Abbot, D. S. (2016). Differences in water vapor radiative transfer among 1D models can significantly affect the inner edge of the habitable zone. *The Astrophysical Journal*, *826*(2), 222.

21. Yang, J., Leconte, J., Wolf, E. T., Merlis, T., Koll, D. D., Forget, F., & Abbot, D. S. (2019). Simulations of Water Vapor and Clouds on Rapidly Rotating and Tidally Locked Planets: A 3D Model Intercomparison. The Astrophysical Journal, 875(1), 46.

---

## Author Comment (AC2) · 1 May 2021

Dear Editor and Referees,

Thank you very much for handling the review on our manuscript "Examining the role of varying surface pressure in the climate of early Earth" (No. cp-2020-55). Your comments have been very helpful for improving the manuscript. In the following, we present replies (in black) to your comments (in blue). Following your comments and suggestions, we have made improvements in the revised manuscript (in red face; we will submit the revised manuscript soon if applicable).

Sincerely,

Jun Yang and Junyan Xiong,

April 30, 2021

========================================================================

Response to Referee #2 cp-2020-55-RC1:

The paper is well written and presents clearly results concerning solutions to the Faint Young Sun Problem (FYSP). However, I identified a fundamental issue requiring clarification. Indeed, the authors use ClimT model, an Earth system modelling toolkit, and CAM3 (a General Circulation Model) for investigating extreme climate conditions without presenting diagnostics showing the validity of their radiative scheme. For instance, the collision-induced absorption is of great importance to the overall radiative budget in dense atmospheres, but its representation in climate models remains uncertain. If RRTGM (the radiative scheme implemented in ClimT) is a state of the art radiative transfer code (and used in many climate models), that not means that this component is adapted for this specific purpose mainly due to a lack of accurate experimental and theoretical data to explore the early Earth (and especially the surface pressure). This point is not easy to solve which explains why I recommend "rejected" rather than "major revision". If the authors want to solve this issue, the methodology is described in Wolf and Toon 2013 (study also using CAM3). Consequently sections 2.1 and 2.2 should describe the general behavior of the radiative schemes AND sets of results demonstrating the validation.

Reply: Thank you very much for the comments on radiative transfer module used in this study. Referee #1 also has the same concern. Our reply is as follows: In Yang et al. (2016), they compared the radiative transfer module of CAM3 with other radiative transfer models as well as two line-by-line radiative transfer models (SMART and LBLRTM). The results are shown in Figures A1 and A2 below. These comparisons showed that at low temperatures, the differences among the models are small, but at high temperatures (>320 K), the differences are relatively large. At 280 K, differences in longwave and shortwave radiation fluxes under clear-sky conditions between CAM3 and the two line-by-line radiative-transfer models are around 10 W m$^{-2}$, and at 300 K, it is less than 15 W m$^{-2}$. The upper limit of $CO_2$ amount that CAM3 can well simulate is about 0.1 bar (Pierrehumbert 2005; Abbot et al. 2013); most of our experiments shown in the manuscript are close to or less than this level.

In the study here, most of our simulations have surface temperatures equal to or lower than 310 K (except the 4.0 bar experiments within which the global-mean surface temperature is higher than 310

K, see in Table 1 of the manuscript), so the model CAM3 is roughly suitable for investigating the effects of varying surface pressure, although the radiative transfer module is not as accurate as other general circulation models (such as LMDG and CAM4_Wolf) and the two line-by-line radiation transfer models.

Of course, we agree that a better radiative transfer module should be employed in the faint young sun problem; but, this does not totally negate the value of this manuscript. The accuracy of the model in $H_2O$ and $CO_2$ radiative transfer would not essentially influence the conclusion of this study. Overall, the results of this study are consistent with basic radiative transfer theories and with previous studies such as Wolf and Toon (2013, 2014), as shown in Table 2 of the original manuscript (or Table 3 of the revised manuscript). In the next step of this project, we will use the model CAM4_Wolf or called ExoCAM (developed by Dr. Eric T. Wolf, same as that used in Wolf and Toon (2013, 2014)) and we will compare the results between ExoCAM and CAM3.

[Figure]

Figure A1. Outgoing longwave radiation at the top of the atmosphere for different radiative transfer models. CAM3, CAM4_Wolf, LMDG, and AM2 are the radiation transfer modules used in atmospheric general circulation models; SBDART is an independent radiation transfer model; and SMART and LBLRTM are line-by-line radiation transfer models. The surface temperature is set to be 250, 273, 300, 320, 340, and 360 K. The atmosphere is assumed to Earth-like (1 bar $N_2$, variable $H_2O$, and 376 ppmv $CO_2$). The temperature structures are moist adiabatic profiles overlain by a 200 K isothermal stratosphere. The atmosphere is assumed to be saturated in water vapor (relative humidity is equal to 100%). The volume mixing ratio of water vapor in the stratosphere is set equal to its value at the tropopause. This figure is from Yang et al. (2016).

[Figure]

Figure A2. Upward shortwave flux at the top of the atmosphere (TOA) from different radiation transfer models. The experimental designs are same as those in Figure A1. The incoming stellar radiation at TOA is 340 W m$^{-2}$, and the solar spectra is used in these calculations. This figure is from Yang et al. (2016).

In addition, here is a list of suggestions to improve the manuscript

- line 4 p6. Why the obliquity is set to 0 ?

Reply: For the obliquity, we set it to be zero because we would like to omit the effect of seasonal cycle and because under this the pattern of ocean heat transport can be easily set up. We agree that this is a too simple design, and a higher obliquity (such as 23.5°) is more reasonable. A non-zero obliquity can influence the meridional gradient of solar radiation; for example, more solar radiation in high latitudes and less solar radiation in low latitudes for an obliquity of 23.5° than those for a zero obliquity. This could also influence the atmospheric circulation and clouds. Previous studies showed that the surface climate is warmer under a higher obliquity, due to the effects of ice albedo feedback, lapse rate feedback, and cloud feedback (Mantsis et al. 2011; Linsenmeier et al. 2015; Kilic et al. 2018; Nowajewski et al. 2018; Kang 2019). These suggested that the $CO_2$ and $CH_4$ concentrations for solving the faint young sun problem should be lower than those shown in this study, if a higher obliquity were used in the simulations.

Using the model CAM3, we do one test within which the planetary obliquity is increased from 0 to 23.5°. As shown in Figure A4, the global-mean surface temperature increases from 294.6 to 300.6 K, i.e., a 6.0 K warming in global mean. The trend and the magnitude of the warming are consistent with previous studies such as Linsenmeier et al. (2015). In future simulations, we will consider a non-zero obliquity, as shown in Figure A3 in page 6 of the reply to Referee #1.

[Figure]

Figure A4: Annual- and zonal-mean surface air temperatures in two CAM3 simulations of different planetary obliquities: 0º for the black line and 23.5º for the red line. The background surface pressure is 1.0 bar, ocean heat transport is close to modern value, $CO_2$ partial pressure is 0.04 bar, and $CH_4$ partial pressure is 1 mbar. The global-mean surface temperature is 294.6 K for the black line and 300.6 K for the red line.

Moreover, an aqua-planet with no continent was used in this study because continental fraction during the Archean is likely much less than present and the detailed land-sea distribution was unknown (Flament et al. 2008; Hawkesworth et al. 2019).

- line 3 p5. Citations concerning the Eocene epoch are irrelevant here (to my knowledge the surface pressure is assumed held constant and the load in carbon dioxide does not overcome 1120 ppmv, so very far from values used in the present study)

Reply: Corrected. We deleted this sentence in the revised manuscript.

- p14 section 4.3. The discussion deserves more attention. As summarized by Charnay et al. 2020 (a review paper entitled "Is the FYSP for Earth Solved?") the explanation of a temperate early Earth is not problematic anymore (as illustrated in the table 2 p15). Despite the cooling provided by the decreasing surface pressure (table 1 p5), this section does not conclude if the FYSP becomes more (or very) problematic to solve.

Reply: Thank you very much for pointing this out. We carefully read the paper of Charnay et al. (2020) and Catling and Zahnle (2020). These two papers completely renew our views on the faint young sun problem. (1) Geological constraints on $CO_2$ concentration during the Archean is still in debate, and possible ranges are 3-15 mbar in 2.69 Ga, 3-25 mbar in 2.5 Ga, 24-140 mbar in 2.77 Ga, 22-700 mbar in 2.75 Ga, or 45-140 mbar in 2.46 Ga (Driese et al. 2011; Sheldon 2006; Kanzaki and Murakami 2015). (2) The $CO_2$ concentration required to maintain a temperate climate at ~2.7 Ga is around 40 mbar under 1.0-bar surface pressure and 1 mbar $CH_4$ or less (Table 2 in the original manuscript). Under

0.5-bar surface pressure, the required $CO_2$ approximately doubles or triples, about 100-120 mbar, which is still in the range of geological constraints. (3) In the abstract of the revised manuscript, we clearly present this view: "The latter $CO_2$ concentration is about twice or triple the corresponding value under 1.0-bar surface pressure, but it is still within the range suggested by geological constraints."

- p15 table 2 (Charnay et al. 2013 and Le Hir et al. 2014 both used a mixed-layer ocean (with Ekman transport for Charnay et al. so more complex than a standard mixed-layer model). please correct this point.

Reply: Corrected.

References:

1. Abbot, D. S., Voigt, A., Li, D., Hir, G. L., Pierrehumbert, R. T., Branson, M., ... & B. Koll, D. D. (2013). Robust elements of Snowball Earth atmospheric circulation and oases for life. Journal of Geophysical Research: Atmospheres, 118(12), 6017-6027.

2. Catling, D. C., & Zahnle, K. J. (2020). The Archean atmosphere. Science Advances, 6(9), eaax1420.

3. Charnay, B., Wolf, E. T., Marty, B., & Forget, F. (2020). Is the faint young Sun problem for Earth solved? Space Science Reviews, 216(5), 1-29.

4. Driese, S. G., Jirsa, M. A., Ren, M., Brantley, S. L., Sheldon, N. D., Parker, D., & Schmitz, M. (2011). Neoarchean paleoweathering of tonalite and metabasalt: Implications for reconstructions of 2.69 Ga early terrestrial ecosystems and paleoatmospheric chemistry. Precambrian Research, 189(1-2), 1-17.

5. Flament, N., Coltice, N., & Rey, P. F. (2008). A case for late-Archaean continental emergence from thermal evolution models and hypsometry. Earth and Planetary Science Letters, 275(3-4), 326-336.

6. Goldblatt, C., Claire, M. W., Lenton, T. M., Matthews, A. J., Watson, A. J., & Zahnle, K. J. (2009). Nitrogen-enhanced greenhouse warming on early Earth. Nature Geoscience, 2(12), 891-896.

7. Hawkesworth, C., Cawood, P. A., & Dhuime, B. (2019). Rates of generation and growth of the continental crust. Geoscience Frontiers, 10(1), 165-173.

8. Kang, W. (2019). Mechanisms Leading to a Warmer Climate on High-obliquity Planets. The Astrophysical Journal Letters, 876(1), L1.

9. Kanzaki, Y., & Murakami, T. (2015). Estimates of atmospheric $CO_2$ in the Neoarchean–Paleoproterozoic from paleosols. Geochimica et Cosmochimica Acta, 159, 190-219.

10. Kilic, C., Lunkeit, F., Raible, C. C., & Stocker, T. F. (2018). Stable equatorial ice belts at high obliquity in a coupled atmosphere–ocean model. The Astrophysical Journal, 864(2), 106.

11. Linsenmeier, M., Pascale, S., & Lucarini, V. (2015). Climate of Earth-like planets with high obliquity and eccentric orbits: implications for habitability conditions. Planetary and Space Science, 105, 43-59.

12. Mantsis, D. F., Clement, A. C., Broccoli, A. J., & Erb, M. P. (2011). Climate feedbacks in response to changes in obliquity. Journal of Climate, 24(11), 2830-2845.

13. Nowajewski, P., Rojas, M., Rojo, P., & Kimeswenger, S. (2018). Atmospheric dynamics and habitability range in Earth-like aquaplanets obliquity simulations. Icarus, 305, 84-90.

14. Pierrehumbert, R. T. (2005). Climate dynamics of a hard snowball Earth. Journal of Geophysical Research: Atmospheres, 110(D1).

15. Sheldon, N. D. (2006). Precambrian paleosols and atmospheric $CO_2$ levels. Precambrian Research, 147(1-2), 148-155.

16. Wolf, E. T., & Toon, O. B. (2013). Hospitable Archean climates simulated by a general circulation model. Astrobiology, 13(7), 656-673.

17. Wolf, E. T., & Toon, O. B. (2014). Controls on the Archean climate system investigated with a global climate model. Astrobiology, 14(3), 241-253.

18. Yang, J., Leconte, J., Wolf, E. T., Goldblatt, C., Feldl, N., Merlis, T., Wang, Y., Koll, D. D. B., Ding, F., Forget, F. & Abbot, D. S. (2016). Differences in water vapor radiative transfer among 1D models can significantly affect the inner edge of the habitable zone. *The Astrophysical Journal*, *826*(2), 222.

---

## Author Comment (AC3) · 1 May 2021

Dear Editor and Referee,

Thank you very much for handling the review on our manuscript "Examining the role of varying surface pressure in the climate of early Earth" (No. cp-2020-55). Your comments have been very helpful for improving the manuscript. In the following, we present replies (in black) to your comments (in blue). Following your comments and suggestions, we have made improvements in the revised manuscript (in red face; we will submit the revised manuscript soon if applicable).

Sincerely,

Jun Yang and Junyan Xiong,

April 30, 2021

================================================================================

Response to Referee #3 cp-2020-55-RC2:

Focusing on the role of atmospheric pressure, Xiong and Yang present one possible solution for the faint young sun paradox. Overall, I think this contribution can move forward our understanding and I think it should eventually be published. However, I have several concerns and questions that I think should be addressed before publication (outlined below).

Reply: Thanks for comments and suggestions. Below, we reply to your comments point-to-point.

Major issues:

At page 2, line 21: First off, there is no geochemical proxy on atmospheric methane, so we simply don't know their upper or lower limit in the past. Second, Pavlov et al. (2001) on Archean kerogens didn't give an upper limit on methane concentration after their modelling exercise. Third, even if they did, many of the Archean kerogens are now believed to be contaminated by the oil drilling, therefore became an unreliable indicator for $CO_2$/$CH_4$ ratio.

Reply: We agree the referee's view on the concentration of $CH_4$. In the revised manuscript, we change the statement to: "Photochemical-ecosystem simulations suggested that $CH_4$ concentration should have been ranged from 100 to 35000 ppmv (Kharecha et al. 2005) or may be higher than 20 ppmv in > 2.4 Ga (Zahnle et al. 2006) or 5000 ppmv in ~3.5 Ga (Zahnle et al. 2019; Catling and Zahnle 2020). In our simulations, we choose an intermediate value, 1 mbar, being equal to ~1000 ppmv under a surface pressure of 1.0 bar or ~2000 ppmv under 0.5 bar."

At page 4, section 2.1: A question to the authors: the 1-D radiative transfer model also has Rayleigh scattering induced changes in planetary albedo, which then linked to the outgoing solar radiation. Did the albedo from 3-D model then coupled with the albedo parameter in the 1-D model? If not, why?

Reply: Yes, both the 1-D and 3-D models have included the effect of Rayleigh scattering and it can influence the planetary albedo. In this aspect, the two models are the same. The main differences between the two models are that the 3-D model includes the feedbacks associated with surface ice, water vapor, and clouds, and the effect of large-scale circulations (such as meridional heat transport) is also considered in the 3-D model but not in the 1-D model.

At page 15, table 2: Even if the authors can ignore the Archean high obliquity hypothesis, why is the obliquity is set to zero? Some justification is needed. Also, if ocean heat transport is a major parameter that differs from previous modeling work, what are the reasons the authors had in choosing their parameter space? Please provide more justification on the benefits of the utilized model and note how it compares to other models.

Reply: For the obliquity, referee #2 has the same comment, and our reply is as follows. For the obliquity, we set it to be zero because we would like to omit the effect of seasonal cycle and because under this the pattern of ocean heat transport can be easily set up. We agree that this is a too simple design, and a higher obliquity (such as 23.5°) is more reasonable. A non-zero obliquity can influence the meridional gradient of solar radiation; for example, more solar radiation in high latitudes and less solar radiation in low latitudes for an obliquity of 23.5° than those for a zero obliquity. This could also influence the atmospheric circulation and clouds. Previous studies showed that the surface climate is warmer under a higher obliquity, due to the effects of ice albedo feedback, lapse rate feedback, and cloud feedback (Mantsis et al. 2011; Linsenmeier et al. 2015; Kilic et al. 2018; Nowajewski et al. 2018; Kang 2019). These suggested that the $CO_2$ and $CH_4$ concentrations for solving the faint young sun problem should be lower than those shown in this study, if a higher obliquity were used in the simulations.

Using the model CAM3, we do one test within which the planetary obliquity is increased from 0 to 23.5°. As shown in Figure A4, the global-mean surface temperature increases from 294.6 to 300.6 K, i.e., a 6.0 K warming in global mean. The trend and the magnitude of the warming are consistent with previous studies such as Linsenmeier et al. (2015). In future simulations, we will consider a non-zero obliquity, as shown in Figure A3 in page 6 of the reply to Referee #1.

[Figure]

Figure A4: Annual- and zonal-mean surface air temperatures in two CAM3 simulations of different

planetary obliquities: 0° for the black line and 23.5° for the red line. The background surface pressure is 1.0 bar, ocean heat transport is close to modern value, $CO_2$ partial pressure is 0.04 bar, and $CH_4$ partial pressure is 1 mbar. The global-mean surface temperature is 294.6 K for the black line and 300.6 K for the red line.

For the magnitude of the ocean heat transport, a lower limit of 0.5 modern value and an upper limit of 2.0 are used in this study. The lower limit is based on the study of snowball Earth initiation: during the ice edge moves closer to the tropics, the ocean heat transport is close to about 0.5 of modern value (Poulsen and Jacob 2004). The upper limit is based on the study of Olson et al. (2020), who employed the ocean heat transport under various parameters. They found that the wind stress has an upper limit of about 2 times the modern value when varying the air pressure from 0.25 to 10 bars (Figure 7 in Olson et al. 2020), so we speculated that the ocean heat transport cannot be much larger than the modern value when varying the air pressure. The real range of the ocean heat transport is required to be examined using fully coupled atmosphere-ocean models; we are doing this kind of simulations and the preliminary result is shown in Figure A3 in page 6 of the reply to Referee #1.

At page 5, line 14-15: even if pCH4 can be set as high as 1E-3 as a modelling exercise, I wonder why the authors didn't mention the concurrent hydrogen flux (or the lack thereof), which according to Kharecha et al., 2005 Geobiology paper, is quantitatively similar to the methane concentration (on a related note, the lead author from the same research group believe the methane estimate in their Kharecha et al. 2005 paper is more reliable than their Pavlov et al. (2001) paper, on top of my major issue 1). Since this article is mainly about the effects of pressure, neglecting a major constitute in the Archean atmosphere seems a bit odd to me. Even if hydrogen eventually escape from the atmosphere, it is still a major constituent in the Archean atmosphere if outgassing is continuous. In addition, hydrogen serves as an indirect greenhouse gas that increases the lifetime of methane through scavenging radicals like OH.

Reply: Thank you very much for pointing this out. In the revised manuscript, we add one paragraph to describe the $H_2$ concentration during the Archean and the reason why we have not considered it in the climate simulations: "In theory, collision-induced absorption of thermal radiation by $N_2$-$H_2$ can have a greenhouse effect. This effect, however, is likely very weak during the Archean eon. The studies of Kharecha et al. (2005) and Ozaki et al. (2018) showed that $H_2$ mixing ratio during the Archean is less than $10^{-4}$. The existence of detrital magnetite particles in Archean riverbeds indicates that the partial pressure of $H_2$ was likely less than 0.01 bar (Kadoya and Catling 2019). According to the radiative transfer calculations of Pierrehumbert and Gaidos (2011) and Wordsworth and Pierrehumbert (2013), the warming effect of $N_2$-$H_2$ absorption is efficient only when the $H_2$ mixing ratio is higher than ~0.1 and meanwhile the background air pressure is significantly greater than the present-day level."

Moreover, the model we employed (as well as in many atmospheric general circulation models those not originally developed for deep-past Earth) does not contain the radiative effect of $N_2$-$H_2$. In future, we will use ExoCAM (Wolf and Toon 2014) to examine the effect of $H_2$. The model ExoCAM includes the greenhouse effect of $N_2$-$H_2$.

Minor issues:

At page 1, line 22-27: One fundamental aspect about seawater temperature reconstructions the author didn't mention is that the delta-$^{18}$O value in seawater can change overtime. Recent analysis on iron oxides, a temperature insensitive sedimentary proxy, shows that the seawater delta-$^{18}$O value can increase by 15 permil since the Archean (Galili et al., 2019 Science).

Reply: Thanks for the suggestion. We add one sentence to point out this in the revised manuscript.

At page 2, line 7-13: in the texts above, the authors argued from multiple lines that the Archean seawater temperature was similar or higher than the modern value. If so, why do they argue the higher $pCO_2$ was maintained by a low surface temperature? The authors argument based on silicate weathering feedback seemingly contradict with their own propositions on surface temperature and $pCO_2$. It may be that this section just needs to be rewritten for clarity.

Reply: Thanks for the comment. We have re-written this paragraph: "The most possible solution to the faint young sun problem is large greenhouse effects from $CO_2$ and $CH_4$. Geological constraints on $CO_2$ concentration during the Archean is still in debate, and possible ranges are 3-15 mbar in 2.69 Ga, 3-25 mbar in 2.5 Ga, 24-140 mbar in 2.77 Ga, 22-700 mbar in 2.75 Ga, or 45-140 mbar in 2.46 Ga (Driese et al. 2011; Kanzaki and Murakami 2015; Catling and Zahnle 2020, and references therein). For $CH_4$, Archean S-MIF indicated its concentration was greater than 20 ppmv (Kazaki and Murakami 2015), and fractionation of xenon isotopes suggested its concentration was higher than 5000 ppmv (Zahnle et al. 2019). Photochemical-ecosystem modeling suggested that $CH_4$ concentration could have been ranged from 100 to 35000 ppmv (Kharecha et al. 2005)."

At page 2, line 24-25: it might be better to reference Pavlov and Kasting (2002) Astrobiology paper for Archean $pO_2$. That paper was the original work that provided the most commonly cited upper limit on Archean $pO_2$. Also, 1% PAL of $O_2$ would contradict the modeling decision of not including oxygen and ozone in their bulk atmosphere composition, which also have pressure broadening effect on $CO_2$ and $H_2O$.

Reply: For the Archean eon (4.0 to 2.5 Ga) discussed in this study, $pO_2$ is very low as suggested by Pavlov and Kasting (2002) and summarized in the recent review paper of Calting and Zahnle (2020). In Pavlov and Kasting (2002), they wrote that "We conclude that the atmospheric $O_2$ concentration must have been $< 10^{-5}$ PAL prior to 2.3 Ga." The Great Oxygen Event (GOE) began around 2.4 Ga and ended ~2.1 to 2.0 Ga. During the entire Archean eon, the atmospheric oxygen is less than $10^{-5}$ PAL (Calting and Zahnle 2020). Therefore, it is reasonable to exclude $O_2$ as well as ozone in the Archean simulations.

References:

1. Catling, D. C., & Zahnle, K. J. (2020). The archean atmosphere. Science Advances, 6(9), eaax1420.

2. Driese, S. G., Jirsa, M. A., Ren, M., Brantley, S. L., Sheldon, N. D., Parker, D., & Schmitz, M.

(2011). Neoarchean paleoweathering of tonalite and metabasalt: Implications for reconstructions of 2.69 Ga early terrestrial ecosystems and paleoatmospheric chemistry. Precambrian Research, 189(1-2), 1-17.

3.  Kadoya, S., & Catling, D. C. (2019). Constraints on hydrogen levels in the Archean atmosphere based on detrital magnetite. Geochimica et Cosmochimica Acta, 262, 207-219.

4.  Kang, W. (2019). Mechanisms Leading to a Warmer Climate on High-obliquity Planets. The Astrophysical Journal Letters, 876(1), L1.

5.  Kanzaki, Y., & Murakami, T. (2015). Estimates of atmospheric $CO_2$ in the Neoarchean–Paleoproterozoic from paleosols. Geochimica et Cosmochimica Acta, 159, 190-219.

6.  Kharecha, P., Kasting, J., & Siefert, J. (2005). A coupled atmosphere–ecosystem model of the early Archean Earth. Geobiology, 3(2), 53-76.

7.  Kilic, C., Lunkeit, F., Raible, C. C., & Stocker, T. F. (2018). Stable equatorial ice belts at high obliquity in a coupled atmosphere–ocean model. The Astrophysical Journal, 864(2), 106.

8.  Linsenmeier, M., Pascale, S., & Lucarini, V. (2015). Climate of Earth-like planets with high obliquity and eccentric orbits: implications for habitability conditions. Planetary and Space Science, 105, 43-59.

9.  Mantsis, D. F., Clement, A. C., Broccoli, A. J., & Erb, M. P. (2011). Climate feedbacks in response to changes in obliquity. Journal of Climate, 24(11), 2830-2845.

10. Nowajewski, P., Rojas, M., Rojo, P., & Kimeswenger, S. (2018). Atmospheric dynamics and habitability range in Earth-like aquaplanets obliquity simulations. Icarus, 305, 84-90.

11. Olson, S. L., Jansen, M., & Abbot, D. S. (2020). Oceanographic considerations for exoplanet life detection. The Astrophysical Journal, 895(1), 19.

12. Ozaki, K., Tajika, E., Hong, P. K., Nakagawa, Y., & Reinhard, C. T. (2018). Effects of primitive photosynthesis on Earth's early climate system. Nature Geoscience, 11(1), 55-59.

13. Paradise, A., Fan, B. L., Menou, K., & Lee, C. (2021). Climate diversity in the solar-like habitable zone due to varying background gas pressure. Icarus, 358, 114301.

14. Pavlov, A. A., & Kasting, J. F. (2002). Mass-independent fractionation of sulfur isotopes in Archean sediments: strong evidence for an anoxic Archean atmosphere. Astrobiology, 2(1), 27-41.

15. Pierrehumbert, R., & Gaidos, E. (2011). Hydrogen greenhouse planets beyond the habitable zone. The Astrophysical Journal Letters, 734(1), L13.

16. Poulsen, C. J., & Jacob, R. L. (2004). Factors that inhibit snowball Earth simulation. Paleoceanography, 19(4).

17. Wordsworth, R., & Pierrehumbert, R. (2013). Hydrogen-nitrogen greenhouse warming in Earth's early atmosphere. science, 339(6115), 64-67.

18. Wolf, E. T., & Toon, O. B. (2014). Controls on the Archean climate system investigated with a global climate model. Astrobiology, 14(3), 241-253.

19. Zahnle, K., Claire, M., & Catling, D. (2006). The loss of mass-independent fractionation in sulfur

due to a Palaeoproterozoic collapse of atmospheric methane. Geobiology, 4(4), 271-283.

20. Zahnle, K. J., Gacesa, M., & Catling, D. C. (2019). Strange messenger: A new history of hydrogen on Earth, as told by Xenon. Geochimica et Cosmochimica Acta, 244, 56-85.